# Dynamic metastable polymersomes enable continuous flow manufacturing

Chin Ken Wong [1]✉, Rebecca Y. Lai [1] & Martina H. Stenzel [1]✉

Polymersomes are polymeric analogues of liposomes with exceptional physical and chemical properties. Despite being dubbed as next-generation vesicles since their inception nearly three decades ago, polymersomes have yet to experience translation into the clinical or industrial settings. This is due to a lack of reliable methods to upscale production without compromising control over polymersome properties. Herein we report a continuous flow methodology capable of producing near-monodisperse polymersomes at scale (≥3 g/h) with the possibility of performing downstream polymersome manipulation. Unlike conventional polymersomes, our polymersomes exhibit metastability under ambient conditions, persisting for a lifetime of ca. 7 days, during which polymersome growth occurs until a dynamic equilibrium state is reached. We demonstrate how this metastable state is key to the implementation of downstream processes to manipulate polymersome size and/or shape in the same continuous stream. The methodology operates in a plug-and-play fashion and is applicable to various block copolymers.

Polymersomes are hollow membrane sacs made from block copolymers. They bear morphological resemblance to liposomes (lipid vesicles) but are structurally endowed with the versatility of polymer chemistry. Polymersomes are therefore more customizable, with surface properties[1-3], size[4,5], shape[6-9], topology[10-13], and permeability[14-17] being some features that are routinely manipulated. This, coupled with their ability to accommodate both hydrophilic and hydrophobic cargoes in their core and membrane, respectively, have led to widespread applications across the fields of drug delivery[18-20], synthetic biology[21], and nanoreactors[22,23].

The current go-to method for producing polymersomes on the lab-scale is a batch process known as nanoprecipitation (or the cosolvent method)[7,14,24-27]. The process generally involves (i) the dissolution of a block copolymer in a good solvent, (ii) the addition of a non-solvent specific to the hydrophobic block, followed by (iii) the removal of the good solvent by evaporation or dialysis. The logic behind the process can be explained as follows: First, the use of a good solvent ensures that the block copolymer of interest is molecularly dissolved and devoid of chain entanglement, a pre-requisite for carrying out effective solution self-assembly. Second, the subsequent addition of a non-solvent, typically water, selectively precipitates the hydrophobic block and initiates polymersome formation. Third and lastly, the good solvent is removed, leaving behind an aqueous polymersome solution for use in solvent-free applications.

Although established, the nanoprecipitation approach suffers from several drawbacks including its poor scalability, batch-to-batch variation, and polydispersity. These drawbacks primarily stem from the second nanoprecipitation step, during which the non-solvent is added to the system, typically either slowly using a syringe pump or abruptly in one portion using a pipette. Irrespective of how the non-solvent is introduced or how vigorously the solution is stirred during the addition process, the timescale of mixing is restricted to the order of seconds. The subjected block copolymer consequently experiences an inhomogeneous change in solvent quality during mixing, triggering nucleation-growth in a non-homogeneous manner. Worse yet, this issue becomes amplified by mass transfer effects when the process is carried out at scales beyond those typical for nanoprecipitation (typically <10 mL).

To negate the effects of batch mixing, researchers have turned to flow-based systems such as microfluidics. A reliable microfluidics approach is the double emulsion templating method[28-30], which relies on the use of flow-focusing chips to confine and self-assemble block copolymers in the oil phase of water/oil/water (w/o/w) double emulsion droplets. Although the approach generates monodisperse

[1]School of Chemistry, University of New South Wales (UNSW), Sydney, NSW 2052, Australia. ✉e-mail: c.kenwong@unsw.edu.au; m.stenzel@unsw.edu.au

polymersomes with high reproducibility, it is somewhat limited in terms of accessible polymersome size (tens to hundreds of μm) and production scalability because the devices used typically only operate at flow rates of only several μL/min. Other microfluidic devices based on flow-focusing (e.g., laminar mixing[31–33] and plug flow mixing)[34,35] enable access to smaller, sub-micron polymersomes, but they tend to compromise polydispersity and likewise have poor scalability owing to their low tolerable flow rates. Another flow-based approach worth noting here is the application of a concurrent polymerization/self-assembly process known as polymerization-induced self-assembly (PISA) process under continuous flow conditions[36,37]. The approach is scalable, but has limited compatibility with certain block copolymer chemistries and solvents, and also typically requires iterative screening and optimization to access a pure polymersome phase.

An alternative flow-based approach sometimes referred to as flash nanoprecipitation[38], relies on the use of miniaturized mixing chambers (micromixers) that reduce the mixing timescale between two incoming solution streams down to the millisecond regime. By employing a micromixer for nanoprecipitation, as opposed to simply conducting nanoprecipitation under batch conditions, one can effectively enhance the uniformity of an overall block copolymer self-assembly process to generate polymersomes in a highly reproducible manner[39–43]. Although proven effective, most reports on this approach employ the use of micromixers with complex internal geometries that are difficult and expensive to manufacture. Furthermore, polymersomes that are generated in this way have always been reported to be kinetically trapped structures[39–42]. From a manufacturing perspective, these polymersomes' kinetically trapped state limits the full potential of a flow setup, whose key benefit lies in its modularity, where one can e.g., equip additional flow accessories in a plug-and-play fashion to integrate and operate downstream processes.

Herein, we lay the groundwork for developing continuous flow approaches towards polymersome manufacturing (i.e., polymersome production with the possibility of downstream manipulation) using affordable, commercially available flow components. The key to our strategy lies in the discovery of a peculiar flow self-assembly condition, which circumvents kinetic traps to produce metastable polymersomes. Under ambient conditions, the lifetime of this metastable state measures ca. 7 days, during which the polymersomes gradually grow in size as they transition into a dynamic equilibrium state. We show that the existence of this metastable state is central to the implementation of downstream processes. As proof of concept, we describe two flow setups capable of downstream polymersome size control (with sub-40 nm precision) and/or shape control. Other highlights of this work include (i) our unparalleled polymersome production rate of, but not limited to, ≥3 grams per hour (g/h)–far surpassing the capabilities of batch polymersome formation methods (including nanoprecipitation), which typically operate at production rates of only several-to-tens of milligrams per hour (mg/h), (ii) the general applicability of the self-assembly process towards various block copolymer types (e.g., diblock, triblock and different corona chemistries), and (iii) the modularity of the flow setup to operate in a plug-and-play fashion.

## Results
### Polymer synthesis and self-assembly
The model polymer used herein is a diblock copolymer, polyethylene oxide-*block*-polystyrene (PEO$_{44}$-*b*-PS$_{86}$), synthesized by reversible addition fragmentation chain transfer (RAFT) polymerization ($M_{n,NMR}$ = 11,260 g/mol, Đ = 1.08). Details on the polymer synthesis and its characterization data are provided in the Supplementary Information (see also Supplementary Fig. 1). This diblock copolymer was selected because it is a common building block for polymersome formation[7,25]. The first iteration of the continuous flow setup used in this work consists of two syringe pumps, a static mixing tee

(micromixer), an equilibration loop, and a collection outlet. A photograph of the setup is shown in Supplementary Fig. 2.

To perform continuous flow self-assembly, we equip the syringe pumps with two separate syringes; one containing the organic phase (1 mg/mL of PEO$_{44}$-*b*-PS$_{86}$ in 2:8 (v/v) 1,4-dioxane/tetrahydrofuran (THF)), and the other containing water. The organic solvent of choice is based on an earlier report on the batch self-assembly of PEO-*b*-PS polymersomes[44]. Both solutions were degassed thoroughly prior to the experiment to minimize outgassing or bubble formation in the setup (see Supplementary Fig. 3 for the degassing procedure). Note that syringe size, solution volume, and length of the equilibration loop do not influence the self-assembly process. They can therefore be adjusted to suit the user's needs (e.g., production scale) or mechanical limitations of their syringe pump.

As a starting point, we investigated the effect of the flow rate ratio on self-assembly. For this, we adjusted the ratio of the flow rate of the organic phase ($Q_{organic}$) and water ($Q_{water}$) asymmetrically while maintaining a total flow rate of $Q_{total}$ = 1 mL/min. For simplicity, we discuss our data below in terms of $Q_{organic}/Q_{total}$, noting that any increments in $Q_{organic}$ must be accompanied by a concomitant decrease in $Q_{water}$.

We performed the self-assembly process at 7 different asymmetric flow rates ranging from $Q_{organic}/Q_{total}$ = 0.1–0.7 (in 0.1 increments). In every case, the product was collected directly into a quartz cuvette and immediately analyzed by dynamic light scattering (DLS). The resulting particle size distributions are shown in Fig. 1b. Each sample's intensity-averaged hydrodynamic diameter ($D_{h,intensity}$) and polydispersity index (PDI) are plotted in Fig. 1c. A summary of the DLS data is further provided in Supplementary Table 1.

All 7 asymmetric flow rates resulted in monomodal particle size distributions with relatively low PDIs of <0.16 (Fig. 1b, c and Supplementary Table 1). At $Q_{organic}/Q_{total}$ ≤ 0.2, minimal changes in particle size were observed. Increments above this value, however, resulted in a linear increase in particle size (see $D_{h,intensity}$ datapoints for $Q_{organic}/Q_{total}$ = 0.3–0.7 in Fig. 1c). We note here that asymmetric flow rates of $Q_{organic}/Q_{total}$ > 0.7 were also tested, but these flow conditions did not result in any particle formation because PEO$_{44}$-*b*-PS$_{86}$ remains molecularly dissolved when the organic solvent content exceeds 70 vol%.

Next, we used transmission electron microscopy (TEM) to probe particle morphology. Shown in Fig. 1d–f are three TEM images of particles produced at $Q_{organic}/Q_{total}$ = 0.2, 0.4, and 0.6, respectively. The gradual increase in $Q_{organic}/Q_{total}$ generated a morphological transition from micelles (Fig. 1d) to a mixed phase of micelles/polymersomes (Fig. 1e), and finally to polymersomes (Fig. 1f). For clarity, the three accessible morphological phases are highlighted in Fig. 1c using different shades of gray.

The self-assembly processes can be explained as follows. At low $Q_{organic}/Q_{total}$ (≤ 0.2), the organic phase containing PEO$_{44}$-*b*-PS$_{86}$ is flowed at low flow rates ($Q_{organic}$ ≤ 0.2 mL/min) and mixed with water, which is conversely flowed at much higher flow rates ($Q_{water}$ ≥ 0.8 mL/min). For all datapoints collected at $Q_{organic}/Q_{total}$ ≤ 0.2 (refer back to Fig. 1b, c), the organic phase is therefore asymmetrically mixed with a large volume excess (specifically, between 4- to 9-folds) of water. Since water is a non-solvent for the PS block, the applied flow conditions trigger PEO$_{44}$-*b*-PS$_{86}$ to undergo phase separation immediately upon contact with water[7,25]. Spherical micelles (Fig. 1d) with a PS core and PEO corona are formed as a result to minimize any unfavorable contact between the PS chains and water.

At 0.2 ≤ $Q_{organic}/Q_{total}$ < 0.6, we approach near-symmetric flow rates, which yield a mixture of micelles and polymersomes (n.b., the latter dominates in population as $Q_{organic}/Q_{total}$ approaches 0.6). Here, the onset of polymersome formation can be ascribed to the increase in $Q_{organic}$ and concomitant decrease in $Q_{water}$. Under these flow conditions, there is an overall improvement in solvent quality, which causes the PS chains to become more swollen and stretched. This leads to an

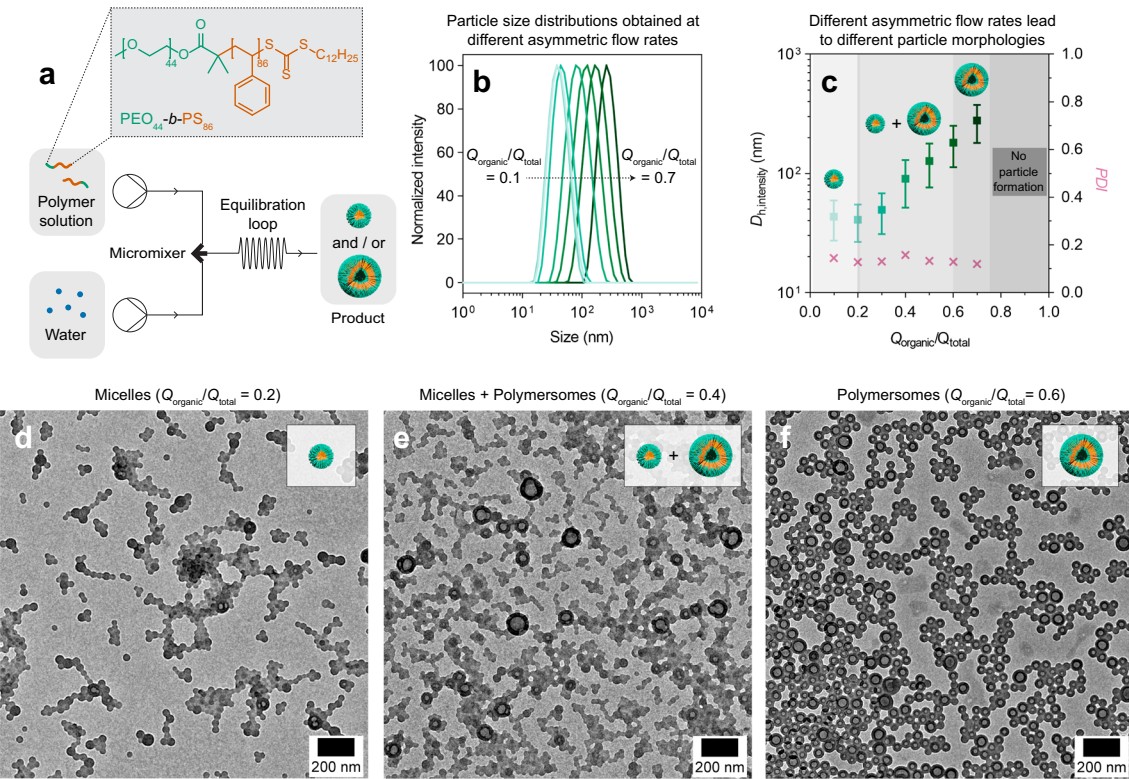

**Fig. 1 | Continuous flow self-assembly. a** Schematic of the continuous flow setup and chemical structure of the polymer (PEO$_{44}$-$b$-PS$_{86}$) used in this work. **b** DLS particle size distributions were obtained at different asymmetric flow rates ($Q_{organic}/Q_{total}$). **c** Intensity-averaged hydrodynamic diameters ($D_{h, intensity}$) expressed in mean ± SD ($n = 3$) and polydispersity indices (PDI) derived from the data shown in (**b**). The different shades of gray in (**c**) depict a pseudo-phase diagram. TEM images of (**d**) micelles obtained at $Q_{organic}/Q_{total} = 0.2$, (**e**) a mixture of micelles and polymersomes at $Q_{organic}/Q_{water} = 0.4$, and (**f**) polymersomes obtained at $Q_{organic}/Q_{total} = 0.6$. All samples were analyzed in their respective organic solvent/water mixtures.

increase in geometric packing parameter, favoring the formation of polymersomes over micelles[45]. We infer, however, based on the co-existence of micelles and polymersomes (see pseudo-phase diagram Fig. 1c) that not all PS chains adopt the same swollen/stretched chain conformation under the flow conditions of $0.2 \leq Q_{organic}/Q_{total} < 0.6$.

Finally, flow conditions between $0.6 \leq Q_{organic}/Q_{total} \leq 0.7$ lead to the formation of a pure phase of polymersomes. As mentioned above, polymersome formation is favored over micelle formation when PS chains adopt a swollen and stretched chain conformation. Under these flow conditions, polymersomes exclusively form because the bulk of PS chains in solution are well-solvated owing to the high organic solvent content present (60–70 vol%).

To summarize our discussions above, we propose a free energy diagram in Fig. 2a to illustrate the two possible self-assembly pathways, namely Pathway 1 (micelle formation) and Pathway 2 (polymersome formation). We hypothesize that micelles formed via Pathway 1 at $Q_{organic}/Q_{total} \leq 0.2$ are kinetically trapped structures, whose morphological transformation is hindered by an activation energy, $E_{A1} \gg k_{B}T$. Polymersomes on the other hand, which are formed via Pathway 2 exclusively at $0.6 \leq Q_{organic}/Q_{total} \leq 0.7$, are more thermodynamically favorable structures occupying a lower free energy minimum. We note here, however, that this is simply a local minimum ($E_{A2} \approx k_{B}T$) in the free energy landscape. Despite being thermodynamically favored, the polymersome structures are in fact metastable (non-equilibrium) structures, as will be discussed shortly.

To prove our hypothesis that the (i) micelles are kinetically trapped structures and (ii) polymersomes are metastable structures, we prepared two fresh batches of micelles and polymersomes at $Q_{organic}/Q_{total} = 0.1$ and 0.7, respectively. The samples were sealed and allowed to age at room temperature for 14 days, during which aliquots were periodically removed for turbidity analysis.

As shown in Fig. 2b(i), we observed no turbidity changes in the micelle sample over the 14-day period. Since turbidity is a function of particle size (due to scattering of light), the data suggest that the micelles maintain a constant particle size throughout the experiment. We further verified this trend by repeating the same experiment, but this time aliquoting the samples for DLS analysis instead. As expected, no changes in micelle size were observed (Supplementary Fig. 4). The data is thus consistent with our hypothesis that the micelles are kinetically trapped structures. Evidently, thermal energy $k_{B}T$ at room temperature is insufficient to overcome the activation energy barrier $E_{A1}$ in the free energy landscape (refer back to Fig. 2a). The system hence remains in a micellar state irrespective of the aging period.

In contrast, the polymersome sample became increasingly turbid in the first 7 days, after which its turbidity plateaus (Fig. 2b(ii)), indicating that they are metastable with a lifetime of ca. 7 day. During this time, the polymersomes undergo a growth process, which we further substantiate by DLS analysis (Supplementary Fig. 4). This gradual increase in polymersome size occurs because thermal energy $k_{B}T$ at room temperature is high enough ($E_{A2} \approx k_{B}T$) to push the system out of its metastable state into a more thermodynamically favorable state (Fig. 2a). The process is also likely facilitated by the high chain mobility of PEO$_{44}$-$b$-PS$_{86}$ within the polymersome membrane (n.b., recall that the organic solvent content in this sample is as high as 70 vol% as it was produced using $Q_{organic}/Q_{total} = 0.7$). The plateau occurs after ca. 7 days as the aged polymersomes eventually occupy a global free energy minimum and are thus considered to be in an equilibrium state.

**Polymersome metastability enables size control**
Next, we leveraged our understanding of the polymersomes' metastability to explore size control in water. Given the high glass transition temperature of PS ($T_{g,PS} \approx 100\,°C$), it should in theory be possible to

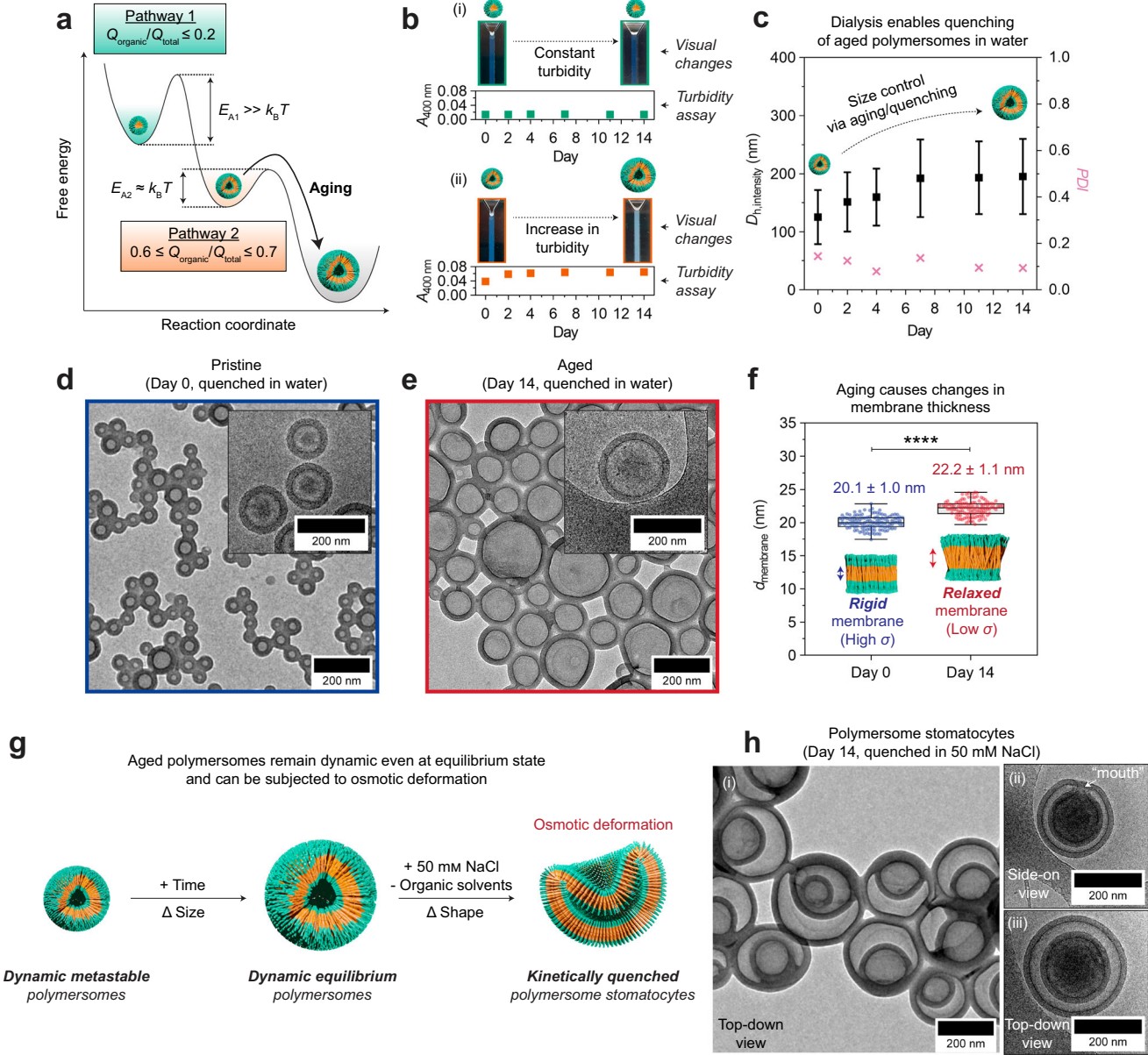

**Fig. 2 | Polymersome metastability and characterization. a** Proposed free energy diagram depicting two accessible continuous flow self-assembly pathways using our methodology. Micelles generated via Pathway 1 are kinetically trapped, while polymersomes generated via Pathway 2 are metastable and grow with age until an equilibrium state is reached. **b** Turbidity changes in a micelle solution ($Q_{organic}/Q_{total} = 0.1$) *vs.* a polymersome solution ($Q_{organic}/Q_{total} = 0.7$) monitored over a 14-day period. Both samples were analyzed in their respective organic solvent/water mixtures. **c** DLS data shows how polymersomes can be quenched in water via dialysis on different aging days to give different polymersome sizes. Intensity-averaged data are expressed in mean ± SD ($n = 3$). TEM images of (**d**) pristine polymersomes quenched on day 0 (immediately after continuous flow self-

assembly) and (**e**) aged polymersomes quenched after 14 days of aging. Corresponding cryo-TEM images are shown inset in (**d**, **e**). **f** Box-and-whiskers plots showing the membrane thickness ($d_{membrane}$) difference (mean ± SD) between the two polymersome samples shown in D and E ($n = 115$ polymersomes based on cryo-TEM images, $P < 0.0001$ determined by paired $t$-test). **g** Schematic illustrating how small metastable polymersomes spontaneously age into larger equilibrium polymersomes. Despite their equilibrium state, the larger aged polymersomes remain dynamic, and can be transformed into polymersome stomatocytes by osmotic deformation. **h** (i) TEM and (ii, iii) cryo-TEM images of stomatocytes obtained by subjecting a 14-day aged polymersome sample to osmotic deformation using 50 mM NaCl.

quench the metastable polymersome structures in different kinetically arrested states at any point during the 7-day growth process. This quenching process should be possible by removing the organic solvent in the system, as the absence of any plasticizing solvent molecules would cause the polymersomes' membrane structure to transition into a glassy quenched state, thereby preventing any further chain rearrangement events.

We verified the above by first checking the stability of the polymersomes in neat water. For this, we prepared a fresh batch of metastable polymersomes (herein referred to as the parent solution),

removed an aliquot immediately after self-assembly, and dialyzed the aliquot against water to remove the organic solvents. The resulting aqueous sample was then probed by DLS (Fig. 2c) and TEM (Fig. 2d) to confirm that the polymersome morphology is retained even in water. DLS showed a size decrease (Supplementary Table 2) upon solvent transfer from the initial organic solvent/water mixture to water, but this is expected because polymersomes are known to swell in the presence of organic solvents[6]. Following removal of the organic solvents, the polymersomes cease to grow with age, as they adopt a frozen (quenched) state in water.

To generate aqueous polymersomes with different sizes, we simply allow the parent solution to age for a different number of days and repeat the same aliquoting/dialysis procedure described in the prior paragraph to quench the structures. The resulting DLS data for six aqueous polymersome samples quenched on days 0, 2, 4, 7, 11, and 14 are shown in Fig. 2c (see Supplementary Table 2 for tabulated DLS data). Consistent with what was observed in Fig. 2b, we observed (i) a linear increase in size for polymersomes quenched between 0–7 days of aging, and (ii) no noticeable size change from day 7 onwards. Collectively, the data prove that size control can indeed be realized in water by exploiting polymersome metastability, and its corresponding lifetime, through the use of a quenching procedure. Our ability to control mean polymersome size with sub-40 nm precision (Fig. 2c and Supplementary Table 2) is a feat inconceivable with conventional polymersomes formation methods.

Next, we provide in Fig. 2d, e two (cryo-)TEM images to highlight the size difference between polymersomes quenched on days 0 and 14, respectively. We determined the membrane thickness ($d_{membrane}$) of both samples by performing statistical analyses on $n = 115$ polymersomes imaged under cryo-conditions. From this, we measured a difference in $d_{membrane}$ of ~2.1 nm (Fig. 2f), which is statistically significant with $P < 0.0001$ based on a paired $t$-test. This indicated to us that the polymersome membrane structure undergoes some form of conformational change during the aging process, possibly explaining the origins of metastability.

We postulate that the observed phenomenon is driven by a reduction in membrane tension. Since the polymersomes that are generated initially on day 0 are relatively small ($D_{h,intensity} = 121 \pm 44$ nm in the non-swollen state; see again Fig. 2d for TEM image), they exhibit conversely high membrane curvatures ($c = 1/r$, where $r =$ polymersome radius), which lead to the generation of an unfavorable amount of membrane tension ($\delta$) in the structure. When left to age in their as-produced organic solvent/water mixture, the polymersomes grow in order to alleviate membrane tension until an equilibrium size is acquired. As earlier discussed in Fig. 2e, this growth process is also accompanied by an increase in membrane thickness (reflecting a looser packed membrane structure), which presumably helps to further relieve the built-up membrane tension.

Our aged polymersomes remain dynamic even upon reaching an equilibrium size. We prove this by demonstrating the susceptibility of our aged polymersomes to a shape transformation process. For context, it is well-established in the literature that polymersomes can be manipulated into non-spherical shapes by means of osmotic deformation[6,46]. A key requirement for this is a high degree of membrane flexibility (dynamicity)[7], which our polymersomes still conveniently possess even in their aged/equilibrium state.

As evidence, we subjected an unquenched 14-day aged polymersome solution to osmotic deformation by adding an aqueous sodium chloride (NaCl) solution (Fig. 2g). We specifically adjusted the salinity of the polymersome solution to 50 mM NaCl (a concentration regularly used to deform polymersomes by osmotic pressure)[47–49], followed by dialysis to remove the organic solvents. This change in salinity generates an osmotic imbalance between the polymersomes' inner compartment and their surrounding solution, causing a net efflux of solvent molecules out of the polymersomes. This in effect drives a reduction in the polymersomes' internal volume and causes the (initially spherical) polymersomes to deform into indented polymersomes known as stomatocytes (TEM and cryo-TEM images in Fig. 2h).

## Robustness of methodology

In the earlier sections of this manuscript, we only discussed results based on flow self-assembly at a polymer concentration of $c_{polymer} = 1$ mg/mL and total flow rate of $Q_{total} = 1$ mL/min. Although polymersome formation under such flow conditions is already more scalable than under batch conditions[50–53], we wanted to assess the

extent to which our methodology can be scaled. To this end, we began exploring the effect of polymer concentration ($c_{polymer} = 1–9$ mg/mL) on polymersome formation. We kept $Q_{organic}/Q_{total}$ and $Q_{total}$ at 0.7 and 1 mL/min, respectively, to target the polymersome morphology as per the pseudo-phase diagram in Fig. 1c. The polymersomes were quenched in water by dialysis, then analyzed by TEM and DLS (Fig. 3a). Remarkably, the polymersome morphology was retained even at the highest polymer concentration tested ($c_{polymer} = 9$ mg/mL; rightmost TEM image inset, Fig. 3a). DLS revealed a linear increase in polymersome size with polymer concentration, possibly reflecting an increase in aggregation number (i.e., the number of block copolymers that make up a single polymersome).

Next, we investigated the effect of total flow rate ($Q_{total} = 1–8$ mL/min) on polymersome formation. We kept $Q_{organic}/Q_{total}$ and $c_{polymer}$ at 0.7 and 1 mg/mL, respectively, to once again target the polymersome morphology. The DLS data for five resulting quenched aqueous polymersome samples are shown in Fig. 3b. As the data show, polymersome size decreases slightly as a function of $Q_{total}$, while PDI values (magenta datapoints in Fig. 3b) decrease exponentially with $Q_{total}$. For polymersomes prepared at the highest total flow rate ($Q_{total} = 8$ mL/min), we measured a notably low PDI of $0.045 \pm 0.015$, indicating near-monodisperse polymersomes (see Fig. 3c for TEM image). We attribute the decreasing size and PDI trends at higher $Q_{total}$ to an increase in flow turbulence during micromixing (Supplementary Fig. 5). The effect of $Q_{total}$, however, diminishes beyond $Q_{total} \geq 8$ mL/min as flow turbulence can no longer be improved beyond the limitations imposed by the geometry of the micromixer.

In the above, we have established that both polymer concentration $c_{polymer}$ and total flow rate $Q_{total}$ can be individually increased to improve the scalability of our methodology. We proceeded to probe their effects in combination. To this end, we conducted the self-assembly process using the following flow conditions: (i) $Q_{organic}/Q_{total} = 0.7$ to target the polymersome morphology, (ii) $c_{polymer} = 9$ mg/mL, and (iii) $Q_{total} = 8$ mL/min. The sample was quenched in water and then analyzed by TEM and DLS (Fig. 3d). Even under these conditions, we observed the formation of a pure phase of polymersomes ($D_{h,intensity} = 121 \pm 39$ nm) with a relatively low PDI of $0.097 \pm 0.015$. What is important to recognize here is that these flow conditions equate to a production rate of $\geq 3$ g of polymersomes/hour, far exceeding the capabilities of typical batch self-assembly processes. The production rate demonstrated herein can undoubtedly be improved by further increasing $c_{polymer}$ and $Q_{total}$, but we did not pursue this as such experiments would require >72 mg of $PEO_{44}$-$b$-$PS_{86}$/minute to conduct.

Next, we examined the applicability of our methodology to other block copolymers. For this, we synthesized two polymers: (i) a triblock terpolymer, $PEO_{44}$-$b$-$P4VP_{21}$-$b$-$PS_{300}$ (see chemical structure in Fig. 3e(i) and Supplementary Fig. 6 for characterization data), which has an additional poly(4-vinylpyridine) (P4VP) middle block and a longer PS block to compensate the change in amphiphilicity incurred by the hydrophilicity of the P4VP block, and (ii) a diblock copolymer, $PAA_{26}$-$b$-$PS_{81}$ (see chemical structure in Fig. 3f(i) and Supplementary Fig. 7 for characterization data), which has a hydrophilic polyacrylic acid (PAA) block to replace the corona-forming block, PEO.

Indeed, as shown in Fig. 3e, f, our methodology is applicable to the two polymers. In both cases, the polymersome morphology can be targeted with only minor alterations in self-assembly conditions (see Supplementary Information for experimental details), which were necessary to either (i) dissolve the longer PS block of $PEO_{44}$-$b$-$P4VP_{21}$-$b$-$PS_{300}$ or (ii) partially protonate the PAA chains of $PAA_{26}$-$b$-$PS_{81}$ (see Experimental Section in the Supplementary Information for details). Despite the changes introduced, we confirmed through aging studies (Fig. 3e, f(ii-iv)) that both polymersome variants exhibit metastability similar to $PEO_{44}$-$b$-$PS_{86}$ polymersomes.

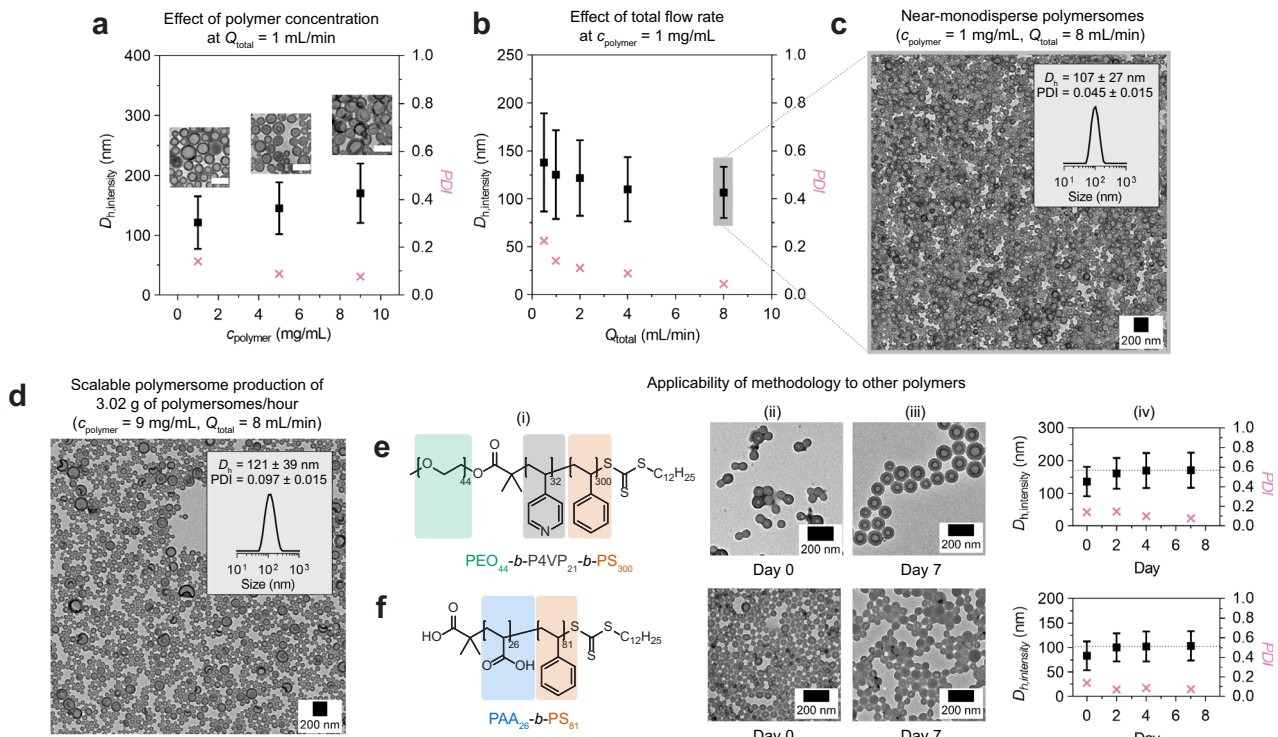

**Fig. 3 | Robustness of methodology. a** DLS data showing the effects of polymer concentration ($c_{polymer}$) on polymersome size. **b** DLS data showing the effects of total flow rate ($Q_{total}$) on polymersome size. **c** TEM image of near-monodisperse polymersomes prepared at $Q_{total}$ = 8 mL/min and $c_{polymer}$ = 1 mg/mL. **d** TEM image of polymersomes produced at a production rate of 3.02 g of polymersomes/hour ($Q_{total}$ = 8 mL/min, $c_{polymer}$ = 9 mg/mL). Shown inset in (**d**) is a DLS particle size distribution for the same sample. All experiments in (**a–d**) were conducted on quenched aqueous polymersome samples originally prepared at $Q_{organic}/Q_{total}$ = 0.7. Chemical structures of (**e**(i)) $PEO_{44}\text{-}b\text{-}P4VP_{21}\text{-}b\text{-}PS_{300}$ and (**f**(i)) $PAA_{26}\text{-}b\text{-}PS_{81}$, and TEM images of their corresponding polymersome structures (ii) before and (iii) after aging for 7 days. The growth process during aging was monitored by (iv) DLS. All intensity-averaged data are expressed in (mean ± SD).

## Continuous flow self-assembly and downstream size control

Based on our understanding of polymersome metastability, we anticipated the possibility of implementing a downstream process in our flow setup to manipulate polymersome size in a similar fashion to our aging experiments (discussed earlier in Fig. 2), but in much shorter reaction times. As proof-of-concept, we extended our flow setup as shown in Fig. 4a (see also Supplementary Fig. 8 for a photograph of the setup) to include an annealing loop and a 100-psi back-pressure regulator down the line. The annealing loop serves as a heated reactor that provides thermal energy to push the polymersomes out of their metastable state and into (more stable) lower free energy states. The back-pressure regulator, on the other hand, increases the boiling point of the organic phase and prevents solvent outgassing/evaporation. The annealing loop used has a volume ($V_{annealing}$) of 2 mL, while the total flow rate ($Q_{total}$) applied of 4 mL/min effectively yields an annealing residence time ($t_{residence/annealing} = V_{annealing}/Q_{total}$) of 30 s. The other flow conditions used were $Q_{organic}/Q_{total}$ = 0.7 and $c_{polymer}$ = 1 mg/mL to target the polymersome morphology. In a typical experiment, we would (i) produce and anneal the polymersomes under flow conditions at different temperatures ($T_{annealing}$ = 20–70 °C), (ii) collect and allow the polymersomes to cool to room temperature, (iii) dialyze them against water to remove the organic solvents, before finally (iv) analyzing the product by DLS and TEM.

As confirmed by DLS analysis (Fig. 4b), polymersome size can indeed be controlled by annealing temperature. TEM analysis (Fig. 4d–f) further verified that the polymersome morphology is maintained across the entire range of annealing temperatures. According to DLS (Fig. 4c), a minimum annealing temperature of 40 °C is required to generate a noticeable increase in polymersome size. At higher temperatures of 40–70 °C, we observe a linear size increment that strongly resembles the linear growth dynamics seen in our earlier

aging study (refer back to day 0–7 DLS datapoints in Fig. 2c). The similarity in upper size limit of the aged (equilibrium) polymersomes and the polymersomes annealed at 70 °C (ca. 190–195 nm in both cases, see final entries of Supplementary Table 2 and Supplementary Table 7 for $D_{h,intensity}$ comparison) imply that we can effectively reduce the time needed for the polymersomes to transition from their metastable state into a global equilibrium state (i.e., from having to age the polymersomes for ca. 7 days down to just 30 s by downstream annealing at 70 °C).

## Continuous flow self-assembly, and downstream size and shape control

To demonstrate the modularity of our approach, we proceeded to implement another downstream process to manipulate polymersome shape, in addition to manipulating polymersome size. For this, we expanded our flow setup to include a cooling loop and a secondary micromixer connected to another syringe pump (Fig. 5a; see Supplementary Fig. 9 for a photograph of the setup). The cooling loop is essential to allow the polymersomes to grow into their equilibrium size following annealing. To clarify this, we show in Fig. 5b a turbidity plot revealing how polymersomes that have been collected after annealing at 70 °C (using the setup in Fig. 4a) require ≥2.5 min of cooling under ambient/batch conditions to achieve their maximum size. In order to translate the growth process under flow conditions, there must be ample time for the annealed polymersomes to grow prior to being subjected to shape transformation.

Although seemingly straightforward, our initial attempts using a cooling loop with a residence time of ca. 7.85 min (i.e., thrice the time required for annealed polymersomes to completely grow under ambient/batch conditions) resulted in barely any growth in polymersome size. This is because, under flow conditions, the heat exchange

**Fig. 4 | Downstream size control. a** Schematic of continuous flow setup used for polymersome self-assembly and downstream annealing to manipulate polymersome size. BPR, backpressure regulator. **b** DLS particle size distributions of aqueous polymersomes prepared at different annealing temperatures ($T_{annealing}$). **c** Intensity-averaged hydrodynamic diameters ($D_{h, intensity}$) expressed in mean ± SD and polydispersity indices (PDI) derived from the data shown in B. TEM images of polymersomes annealed at (**d**) 20 °C, (**e**) 50 °C and (**f**) 70 °C for a residence time under heating ($t_{residence, annealing}$) of 30 s. Flow conditions used for polymersome formation: $Q_{total} = 4$ mL/min, $Q_{organic}/Q_{total} = 0.7$, and $c_{polymer} = 1$ mg/mL. All samples in (**b**–**f**) were dialyzed against water prior to analysis.

between the polymersome solution and its surrounding environment (i.e., air at room temperature) is so efficient that the cooling rate of the polymersome solution exceeds the growth rate of the annealed polymersomes. We found that this can be overcome by placing the cooling loop in an incubator at 40 °C. This reduces the cooling rate sufficiently as to allow the annealed polymersomes to achieve their equilibrium size during the prescribed residence time.

The secondary micromixer, which is placed downstream of the cooling loop (Fig. 5a), serves as a junction for the introduction of an additive (NaCl solution) needed to osmotically deform the annealed/grown polymersomes. In a typical experiment, we would generate a salinity change of 50 mM NaCl by micromixing the annealed/grown polymersome solution with a concentrated NaCl solution (5.05 M) at a flow rate of 4 mL/min and 0.04 mL/min, respectively. We found it crucial to introduce only a small amount of concentrated NaCl solution (as opposed to larger volumes of diluted NaCl solution) as this ensures minimal deviations in solvent quality after micromixing, thus preventing any morphological deviations beyond the intended shape transformation process.

To produce stomatocytes in a continuous fashion, we used the setup in Fig. 5a and applied the following flow conditions: (i) $c_{polymer} = 1$ mg/mL, (ii) $Q_{organic}/Q_{total} = 0.7$, (iii) $Q_{total} = 4$ mL/min, (iv) $T_{annealing} = 70$ °C with $t_{residence/annealing} = 30$ s, (v) $T_{cooling} = 40$ °C with

$t_{residence/cooling} = 7.5$ min, and (vi) $c_{NaCl} = 5.05$ M at $Q_{NaCl} = 0.04$ mL/min. The as-produced solution was then dialyzed to remove the organic solvents, and subsequently solvent exchanged with water to facilitate imaging. Finally, we confirmed the stomatocyte morphology using a combination of TEM and cryo-TEM (Fig. 5c).

## Discussion

In this report, we first detailed the use of a continuous flow setup with commercially available parts to produce dynamic metastable polymersomes. At room temperature, the metastable state has a lifetime of ca. 7 days, after which the polymersomes reach an equilibrium state. Despite being at equilibrium, the polymersome structures remain dynamic and can undergo shape transformation into stomatocytes. Our methodology is robust as it accommodates various flow conditions to, for instance, upscale polymersome production (>3 g/h) or yield polymer micelles. The self-assembly process is not polymer-specific and can be applied to block copolymers with different block chemistries/lengths so long as they are geometrically tailored to target the polymersome morphology. We then expanded our flow setup by applying our knowledge of the polymersomes' dynamic metastable properties. Specifically, we demonstrated that downstream processes such as thermal annealing and/or secondary micromixing can be performed in a continuous fashion to

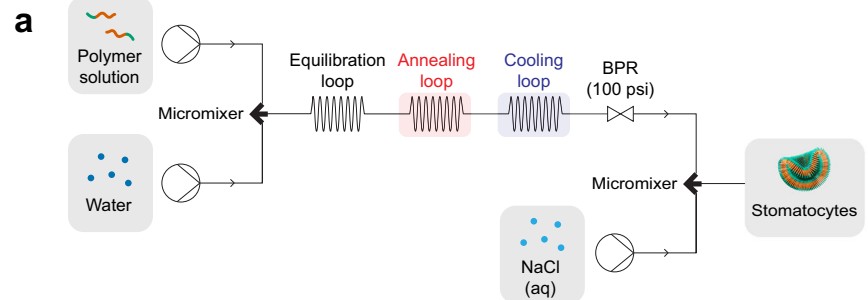

Continuous flow self-assembly and downstream polymersome shape transformation

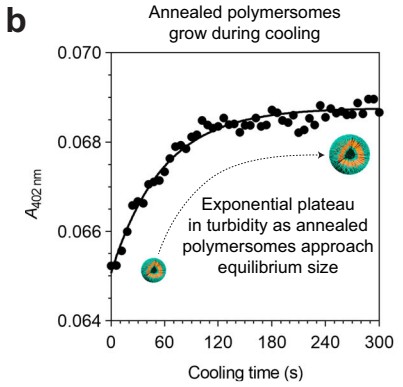

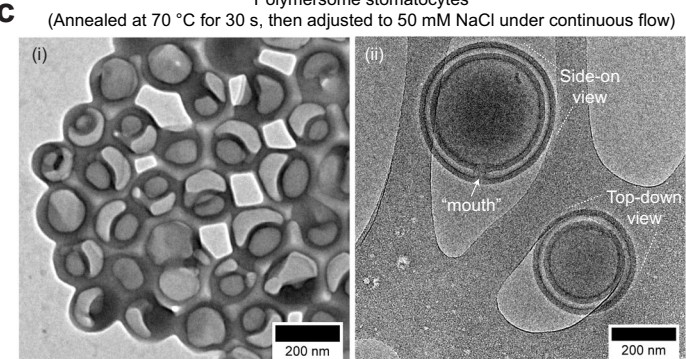

**Fig. 5 | Downstream size control and shape transformation. a** Schematic of continuous flow setup used for sequential polymersome self-assembly, downstream polymersome size control, and shape transformation. BPR, backpressure regulator. **b** Turbidity assay monitored at $\lambda = 402$ nm revealed that the growth process of 70 °C-annealed polymersomes occurs during cooling. **c** (i) TEM and (ii) cryo-TEM images of polymersome stomatocytes obtained by downstream annealing and shape transformation. Flow conditions used for polymersome formation: $Q_{total} = 4$ mL/min, $Q_{organic}/Q_{total} = 0.7$, and $c_{polymer} = 1$ mg/mL. Downstream manipulation conditions: $T_{annealing} = 70$ °C, $t_{residence, annealing} = 30$ s, $T_{cooling} = 40$ °C, $t_{residence, cooling} = 7.85$ min, $c_{NaCl} = 5.05$ M and $Q_{NaCl} = 0.04$ mL/min.

manipulate mean polymersome size and/or polymersome shape. Our work represents a significant contribution to the nanoparticle manufacturing sector, as it provides not only a fundamental basis for the production of near-monodisperse polymersomes at scale but also showcases how polymersome properties can be precisely tuned on the nanoscale, all under flow conditions. Moving forward, we anticipate that other downstream processes such as surface functionalization and in-line purification can be achieved with slight modifications to our reported setup. Finally, in order to accelerate the clinical translation of polymersomes, further advancements in this area should prioritize the development of more clinically relevant polymersomes (e.g., biodegradable/stimuli-responsive polymersomes) to ensure optimal efficacy and safety for patients.

## Data availability

The authors declare that all data supporting the findings of this study are available within the article and its Supplementary Information. Source data are provided with this paper. All other data are available from the corresponding author upon request. Source data are provided with this paper.

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

## Acknowledgements

We acknowledge the use of the Cryo-Electron Microscopy Facility through the Victor Chang Cardiac Research Institute Innovation Centre, funded by the NSW government, and the Electron Microscope Unit and the Nuclear Magnetic Resonance Facility within the Mark Wainwright Analytical Centre (MWAC) at UNSW Sydney. R.Y.L. is grateful for financial support from the UNSW Scientia PhD Scholarship Scheme. This work was supported by the Australian Research Council (ARC) through an ARC Discovery Project (DP200101918) and ARC Laureate Fellowship (FL200100124) to M.H.S.

## Author contributions

C.K.W. conceived the project, designed and performed the experiments, and wrote the manuscript. R.Y.L. performed the experiments on

PAA$_{26}$-*b*-PS$_{81}$ and cryo-TEM imaging. M.H.S acquired funding for the research, provided project supervision, and was involved in the manuscript preparation.

## Competing interests

C.K.W., R.Y.L. and M.H.S. are co-inventors of a PCT application (No. PCT/AU2022/051293) currently assigned to NewSouth Innovations Pty Limited. R.Y.L. and M.H.S. declare no other competing interests. C.K.W. founded Vesiculate to commercialize polymersomes after completion of this study.
