## [Peer Review File · Nature Communications]

Dynamic Metastable Polymersomes Enable Continuous Flow ManufacturingEditorial Note: Parts of this Peer Review File have been redacted as indicated to maintain the confidentiality of unpublished data.

REVIEWER COMMENTS

Reviewer #1 (Remarks to the Author):

The authors have provided a very unique process that can create polymersomes with controllable sizes at much higher throughput than most, if not all, available formulation methods. This work has the potential to be highly impactful in aiding the translation of polymersome technology into clinical trials and beyond. I am very enthusiastic about this paper and really enjoyed reading it. The flow was very logical and experimental evidence is presented for the great majority of claims.

I recommend very minor revisions in accordance with the following remarks:

Robert Prud'homme's group in Princeton has developed a method called Flash Inverse Nanoprecipitation that is capable of high throughput monodisperse nanoparticle production. I think it could be important to compare and contrast benefits associated with the system developed here and this unique system. Furthermore, inverse nanoprecipitation (solvent injection) is capable of forming monodisperse polymersomes, albeit at low concentrations. It could aid in the discussion to include information about this.

The statement made on Page 4 line 14-15 regarding the syringe size, etc having no effect on the self-assembly process should be supported by citations or experiments.

In Figure 1 B it is difficult to distinguish which lines correspond with which concentrations. A color may help here (Similar to Figure 4B).

Figure 2G makes it appear as if osmotic pressure is felt only at a point, when Jan Van Hest's group suggests it is more of an elongation force that ultimately leads to internal collapse of stomatocytes. If it is believed that this force is unidirectional, I think it would be helpful to explain why this is believed and support with citations.

Based on figures and data alone it is difficult to understand precisely where the "sub-40 nm precision" conclusion is coming from. Is this meant to be supported by TEM images? It may be helpful to explain this in more detail on page 10 line 6.

Claim made at the end of the paragraph on page 12 line 13 needs to be supported with citations.

Again, I thoroughly enjoyed this paper and applaud the authors for their very interesting approach to an important translational problem. However, there appears to be a limitation associated with each polymersome having a polystyrene hydrophobic block. PS is not always used in clinical applications, which appear to be dominated by polyesters and other biodegradable blocks. I think it could really increase the impact of the conclusions to discuss potential translation to less hydrophobic or stimuli-responsive blocks.

Reviewer #2 (Remarks to the Author):

Wong et al. describe in this article a continuous flow methodology for production of polymersomes at a relatively large scale (≥ 3 g/h). While the topic of the article is important due to the need of efficient production of nanocarriers for various applications, there are critical issues, which prevent this manuscript for acceptance in Nature Communications. After solving the issues, a revised version will be appropriate to be submitted to a more specialized journal.

General comments:

1. The method presented in the manuscript is based on the combination of a static mixing tee (Y-junction) as a small-scale mixing chamber and a continuous flow setup which reduces the mixing time, while the equilibration loop allows for a good control over the size and shape of polymersomes due to the polymersomes' metastability in the organic solvent/water mixture. However, compared to the current progress of science in the self-assembly process of vesicles formation (polymersomes and giant unilamellar vesicles), the method brings an elegant optimization in one of the polymersome preparation methods however without being a breakthrough in the field.
2. The Introduction does not contain the real state-of-art in the field regarding the polymersomes production because the well known film rehydration method for polymersome formation and loading with molecules is not presented with its advantages both in terms of polymersomes high yield production and encapsulation efficiency. The Introduction should be improved to present all relevant methods for polymersomes formation and their advantages or still open questions in the field.
3. There is a confusion the authors include in the Introduction by considering synthetic giant unilamellar vesicles (GUVs) as polymersomes. In the field there is a clear distinction between the vesicles with sizes in the nanometer range (polymersomes) and the vesicles with micrometer sizes (GUVs), similarly to the notions of liposomes and lipid GUVs. This distinction is based both on the methods of production and the properties of the vesicles membrane (stability, curvature, etc). The focus of this manuscript is in preparation of polymersomes. Therefore, the Introduction should be corrected accordingly to avoid misunderstanding and to clearly indicate the relevance of the method the authors propose.
4. A serious critical aspect is related to the low mechanic stability of polymersomes obtained by the present method, which downgrade them for any application, as they are less stable even than PEGylated-liposomes. Usually polymersomes stability is of several months, depending on the type of amphiphilic copolymer. Therefore, this method should be significantly improved to allow formation of stable polymersomes for longer periods of time than one week, as this is a real bottle neck for further applications.
5. The authors indicate that the method they propose can be used for any type of amphiphilic copolymer. However, they selected as example to test their method for polymersomes formation, PEO-

b-PS, which has specific properties that cannot be extrapolated for other types of amphiphilic copolymers. In addition, both the chemistry of synthesized block copolymers has been well established and the theoretical background of polymer self-assembly via the solvent switch approach has been previously explored in detail, e.g. by the groups of A. Eisenberg, T.P. Lodge or J.C.M. van Hest. To validate this method for a variety of copolymers, the authors should prove it at least for two different types of copolymers in terms of molecular properties.

6. The characterization of polymersomes only by DLS (Pag.4 line 25) is not enough to prove the hollow sphere architecture. Static light scattering experiments should be performed and combined to distinguish whether the spherical nanoobjects are polymersomes.

7. Cryo-TEM will be important to give the necessary details of the polymersome membrane that are not clearly visible from the presented TEM micrographs (Fig. 1). Besides, the resolution of all TEM micrographs should be significantly improved. The change in TEM micrographs as stated, "The three accessible morphological phases are highlighted in Figure 1C using different shades of black" is confusing. It is essential to present the raw data, not highlighted images, to avoid biases.

8. Figures 2 D and E regarding TEM micrographs of polymersomes are different from what we expect to have when hollow sphere architecture is present; the spherical nanoobjects have a darker core that is not specific for polymersomes. Therefore it is not clear that the nanoobjects in these figures are polymersomes. Cryo-TEM will elucidate this issue.

9. Figures 4 D-F indicate aggregation of the spherical nanoobjects, which is a severe limitation for further applications. How the aggregation can be avoided?

Reviewer #3 (Remarks to the Author):

In this manuscript, the authors reported a continuous flow methodology capable of producing near-monodisperse polymersomes at scale (≥ 3 g/h). They also demonstrated downstream processes (thermal annealing and/or secondary micro-mixing) to manipulate polymersome size (with sub-40 nm precision) and/or polymersome shape. This work is meaningful for the production of near-monodisperse polymersomes at scale, as well as the control of polymersomes properties under flow conditions. I would recommend its publication on NC after major revision.

1. The authors claimed that they specifically adjusted the salinity of the polymersome solution to 50 mM NaCl, followed by dialysis to remove the organic solvents. Why choose 50 mM NaCl?

2. At the highest total flow rate ($Q_{total} = 8$ mL/min), a low PDI of 0.045 ± 0.015 was obtained, indicating near-monodisperse polymersomes. The monodispersity is very interesting, and the authors should investigate the formation mechanism of the monodispersity.

3. A production rate of ≥ 3 g of polymersomes/hour was achieved. Further increasing $C_{polymer}$ and Q_{total} , the production rate can be improved. Why the authors did not pursue this?

4. The authors also manipulated polymersome shape by expanding the flow setup to include a cooling loop and a secondary micromixer connected to another syringe pump. It was crucial to introduce only a small amount of concentrated NaCl solution, as this ensures minimal deviations in solvent quality after micromixing, thus preventing any morphological deviations. How to precisely control a small amount of concentrated NaCl solution? What is the exact value and exact quantity? Any preliminary data/experiment?

Black = comments from the reviewers

Blue = our response to the reviewers

Green = reproduced texts from the original/revise manuscript

All additions/changes made to the manuscript are highlighted in yellow to enable tracked changes.

REVIEWER COMMENTS

Reviewer #1 (Remarks to the Author):

The authors have provided a very unique process that can create polymersomes with controllable sizes at much higher throughput than most, if not all, available formulation methods. This work has the potential to be highly impactful in aiding the translation of polymersome technology into clinical trials and beyond. I am very enthusiastic about this paper and really enjoyed reading it. The flow was very logical and experimental evidence is presented for the great majority of claims.

I recommend very minor revisions in accordance with the following remarks:

Comment #1: *Robert Prud'homme's group in Princeton has developed a method called Flash Inverse Nanoprecipitation that is capable of high throughput monodisperse nanoparticle production. I think it could be important to compare and contrast benefits associated with the system developed here and this unique system. Furthermore, inverse nanoprecipitation (solvent injection) is capable of forming monodisperse polymersomes, albeit at low concentrations. It could aid in the discussion to include information about this.*

Response:

We thank the Reviewer for their suggestion.

We have added a reference to Prof. Robert Prud'homme's first paper where he first introduced the term "flash nanoprecipitation". We have also modified our Introduction to clarify that the use of micromixers to conduct nanoprecipitation is sometimes referred to as flash nanoprecipitation in the literature:

"An alternative flow-based approach, sometimes referred to as flash nanoprecipitation,³⁸ relies on the use of miniaturized mixing chambers (micromixers) that reduce the mixing timescale between two incoming solution streams down to the millisecond regime. By employing a micromixer for nanoprecipitation, as opposed to simply conducting nanoprecipitation under batch conditions, one can effectively enhance the uniformity of an overall block copolymer self-assembly process to generate polymersomes in a highly reproducible manner.³⁹⁻⁴² Although proven effective, most reports on this approach employ the use of micromixers with complex internal geometries that are difficult and expensive to manufacture."

Note, however, that we have chosen not to elaborate on *inverse* flash nanoprecipitation specifically because we view the process as somewhat akin to templated layer-by-layer (LbL) self-assembly, as opposed to a true bottom-up self-assembly/nanoprecipitation process such as that seen in *traditional* flash nanoprecipitation. In fact, we support our view by quoting a statement made in a recent paper¹ by Prof. Prud'homme's group where they distinguished nanoparticles made from *inverse* flash nanoprecipitation from polymersomes:

"Called "inverse Flash NanoPrecipitation (iFNP)," the technique achieves biologic loadings (wt% of total formulation) that are 5–15× higher than typical values (9–27% versus < 2%). In contrast to liposomes and polymersomes, we sequentially assemble the polymer layers to form the final nanocarrier".

Comment #2: *The statement made on Page 4 line 14-15 regarding the syringe size, etc having no effect on the self-assembly process should be supported by citations or experiments.*

Response:

We appreciate the suggestion provided by the Reviewer. However, we do not believe that it is necessary to provide citations or conduct experiments to support our statement made on previously on Page 4, Line 14-15 (Page 4, Line 17 in the revised manuscript) for the following reasons:

Firstly, both syringe size and solution volume are simply device-related parameters that a user inputs into a pump program. They should not be considered as experimental parameters because they do not bear any impact on the self-assembly process. To illustrate this, let's consider two scenarios where a 10 mL and 20 mL syringe are used in separate self-assembly experiments. Irrespective of which syringe is used, the self-assembly would remain identical because in both experiments the pumps would have been set to the dispense at identical flow rates. The only notable distinction between the two experiments would be the volume of nanoparticles generated, as the 10 mL syringe has half the capacity of the 20 mL syringe.

Secondly, the length of the equilibration loop likewise has no impact on the self-assembly process. To elaborate, let's again consider two hypothetical experiments: one using a 1 mL equilibration loop and the other using a 2 mL equilibration loop. In both scenarios, the self-assembly process would be identical because the equilibration loop is placed downstream to the micromixer, where the self-assembly process truly occurs. The only real difference between the two experiments is the duration of time the nanoparticles resides in the equilibration loop immediately after self-assembly ($t_{\text{residence}}$). Assuming a total flow rate of 1 mL/min is used, the 1 mL equilibration loop would provide a $t_{\text{residence}} = 1$ min, while the 2 mL equilibration loop would provide $t_{\text{residence}} = 2$ min.

Now, although our polymersomes are metastable in nature, we know from our aging studies (refer back to Figure 2C) that the lifetime of this metastable state ($t_{\text{metastable}}$) is ca. 7 days. Considering the significant difference in timescales between $t_{\text{residence}}$ (min/sec) and $t_{\text{metastable}}$ (~7 days), it is reasonable to claim that the length of the equilibration loop has no impact on the self-assembly process. Of course, one may argue that the equilibration loop can be extended to cover a $t_{\text{residence}}$ of several days, but such an experiment would not be very practical.

Comment #3: *In Figure 1 B it is difficult to distinguish which lines correspond with which concentrations. A color may help here (Similar to Figure 4B).*

Response:

We have updated Figure 1B and Figure 4B to incorporate the Reviewer's suggestions. Both updated figures are reproduced below for clarity.

Updated Figure 1. (A) Schematic of the continuous flow setup and chemical structure of the polymer (PEO₄₄-b-PS₈₆) used in this work. (B) DLS particle size distributions obtained at different asymmetric flow rates ($Q_{\text{organic}}/Q_{\text{total}}$). (C) Intensity-averaged hydrodynamic diameters ($D_{h, \text{intensity}}$) and polydispersity indices (PDI) derived from the data shown in B. The different shades of black in C depict a pseudo-phase diagram. TEM images of (D) micelles obtained at $Q_{\text{organic}}/Q_{\text{total}} = 0.2$, (E) a mixture of micelles and polymersomes at $Q_{\text{organic}}/Q_{\text{water}} = 0.4$ and (F) polymersomes obtained at $Q_{\text{organic}}/Q_{\text{total}} = 0.6$. All samples were analyzed in their respective organic solvent/water mixtures.

Updated Figure 4. (A) Schematic of continuous flow setup used for polymersome self-assembly and downstream annealing to manipulate polymersome size. BPR, backpressure regulator. (B) DLS particle size distributions of aqueous polymersomes prepared at different annealing temperatures ($T_{annealing}$). (C) Intensity-averaged hydrodynamic diameters ($D_{h, intensity}$) and polydispersity indices (PDI) derived from the data shown in B. TEM images of polymersomes annealed at (D) 20 °C, (E) 50 °C and (F) 70 °C for a residence time under heating ($t_{residence, annealing}$) of 30 s. Flow conditions used for polymersome formation: $Q_{total} = 4$ mL/min, $Q_{organic}/Q_{total} = 0.7$ and $c_{polymer} = 1$ mg/mL. All samples in B-F were dialyzed against water prior to analysis.

Comment #4: *Figure 2G makes it appear as if osmotic pressure is felt only at a point, when Jan Van Hest's group suggests it is more of an elongation force that ultimately leads to internal collapse of stomatocytes. If it is believed that this force is unidirectional, I think it would be helpful to explain why this is believed and support with citations.*

Response:

We thank the Reviewer for raising this issue.

The Reviewer is correct in that the osmotic pressure that is applied onto a polymersome structure during shape transformation is not unidirectional. We had initially thought that the addition of the red arrow in Figure 2G would help guide a non-expert reader to envisage how an originally spherical polymersomes can be deformed into a (bowl-like) stomatocyte structure. In hindsight, we admit that this is misleading since the deformation process is not caused by the application of an external force at a single point as we have indicated with the red arrow in Figure 2G. We clarify that we are not suggesting a shape transformation pathway that is any different to what was proposed by the van Hest group in their seminal work.² The deformation process is in fact driven by a reduction in internal volume, which is in turn caused by the rapid efflux of organic solvents from the polymersome core due to osmotic imbalance. Despite the misleading schematic, we had already described the shape transformation mechanism on Page 11, Lines 7-11:

“This change in salinity generates an osmotic imbalance between the polymersomes’ inner compartment and their surrounding solution, causing a net efflux of solvent molecules out of the polymersomes. This in effect drives a reduction in the polymersomes’ internal volume and causes the (initially spherical) polymersomes to deform into indented polymersomes known as stomatocytes (TEM and cryo-TEM images in Figure 2H).”

To prevent further confusion regarding this matter, we have proceeded to remove the red arrow from Figure 2G.

Comment #5: *Based on figures and data alone it is difficult to understand precisely where the "sub-40 nm precision" conclusion is coming from. Is this meant to be supported by TEM images? It may be helpful to explain this in more detail on page 10 line 6.*

Response:

We apologise for the lack of clarity.

We have added a reference to our DLS data in Figure 2C and Table S2 to back our claim made previously on Page 10, Line 6 (Page 10, Line 9 in the revised manuscript). We have also amended the sentence slightly to enhance clarity:

“Our ability to control ~~polymersome size distribution~~ mean polymersome size with sub-40 nm precision (Figure 2C and Table S2) is a feat inconceivable with conventional polymersomes formation methods.”

Comment #6: *Claim made at the end of the paragraph on page 12 line 13 needs to be supported with citations.*

Response:

To support our claim made previously on Page 12, Line 13 (Page 12, Line 11 in the revised manuscript), we have added 4 references to papers published between 2011-2021, where PEO-*b*-PS polymersomes have been prepared by batch nanoprecipitation. The polymersome production rate in each reference is provided in **[bolded brackets]** below.

50. Meeuwissen, S. A., Kim, K. T., Chen, Y., Pochan, D. J. & van Hest, J. C. M. Controlled shape transformation of polymersome stomatocytes. *Angew. Chem. Int. Ed.* **50**, 7070–7073 (2011).
[Polymersome production rate = 3.33 milligrams/hour]
51. Nijemeisland, M., Abdelmohsen, L. K. E. A., Huck, W. T. S., Wilson, D. A. & van Hest, J. C. M. A compartmentalized out-of-equilibrium enzymatic reaction network for sustained autonomous movement. *ACS Cent. Sci.* **2**, 843–849 (2016).
[Polymersome production rate = 6.67 milligrams/hour]
52. Kim, J. & Kim, K. T. Polymersome-Based Modular Nanoreactors with Size-Selective Transmembrane Permeability. *ACS Appl. Mater. Interfaces* **12**, 23502–23513 (2020).
[Polymersome production rate = 20 milligrams/hour]
53. Sun, J., Rijpkema, S. J., Luan, J., Zhang, S. & Wilson, D. A. Generating biomembrane-like local curvature in polymersomes via dynamic polymer insertion. *Nat. Commun.* **12**, 2235 (2021).
[Polymersome production rate = 3.33 milligrams/hour]

For comparison, the highest polymersome production rate we have achieved is 3.02 grams/hour (data shown in Figure 3D in our manuscript).

Comment #7: *Again, I thoroughly enjoyed this paper and applaud the authors for their very interesting approach to an important translational problem. However, there appears to be a limitation associated which each polymersome having a polystyrene hydrophobic block. PS is not always used in clinical applications, which appear to be dominated by polyesters and other biodegradable blocks. I think it could really increase the impact of the conclusions to discuss potential translation to less hydrophobic or stimuli-responsive blocks.*

Response:

We acknowledge the concerns raised by the Reviewer regarding the lack of biological relevance of polystyrene (PS) in clinical applications. Although we share the same sentiment as the Reviewer, it is worthwhile pointing out that, despite being non-biodegradable, PEO-*b*-PS polymersomes still possess value in the medical realms. To elaborate, the Leroux group from ETH Zürich, have for example, recently reported³ on the use of PEO-*b*-PS polymersomes for the oral treatment and diagnosis of hyperammonia, a metabolic condition characterized by an abnormally high level of ammonia in blood, which can, at times, be life-threatening. Prior to publishing this work, the authors of this work had already filed for patent applications worldwide (see e.g., Patent No.: WO2019053578A1, US20200283583A1, EP3668927A1). According to the authors in their *Conflict of Interest* disclosure statement,³ the patents have been licensed to Versantis AG, a clinical-stage pharmaceutical company that focuses on the development of new generation orphan medications.

Having said the above, we do not deter the fact that clinical applications would ultimately benefit from the use of clinically relevant polymersomes. Biodegradable polyesters or biocompatible stimuli-responsive polymers, as suggested by the Reviewer, certainly hold great promise in this regard. We have taken the advice of the Reviewer and expanded our conclusion to emphasize the need for more clinically relevant polymersomes:

“Finally, in order to accelerate the clinical translation of polymersomes, further advancements in this area should prioritize the development of more clinically relevant polymersomes (e.g., biodegradable/stimuli-responsive polymersomes) to ensure optimal efficacy and safety for patients.”

Finally, we extend our gratitude to the Reviewer for their compliment and valuable feedback aimed at improving our manuscript. After nearly dedicating a decade on research on polymersomes, we genuinely believe that this work stands as one of our most significant contributions to the field. We hope that forthcoming readers will likewise recognize and appreciate the impact of our work.

Reviewer #2 (Remarks to the Author):

Wong et al. describe in this article a continuous flow methodology for production of polymersomes at a relatively large scale (≥ 3 g/h). While the topic of the article is important due to the need of efficient production of nanocarriers for various applications, there are critical issues, which prevent this manuscript for acceptance in Nature Communications. After solving the issues, a revised version will be appropriate to be submitted to a more specialized journal.

Response:

We appreciate the Reviewer’s acknowledgement of our work in addressing the pressing need for “*efficient production of nanocarriers for various applications*” despite the presence of certain “*critical issues*”. We have responded to these concerns in detail below.

General comments:

Comment #1: *The method presented in the manuscript is based on the combination of a static mixing tee (Y-junction) as a small-scale mixing chamber and a continuous flow setup which reduces the mixing time, while the equilibration loop allows for a good control over the size and shape of polymersomes due to the polymersomes’ metastability in the organic solvent/water mixture. However, compared to the current progress of science in the self-assembly process of vesicles formation (polymersomes and giant unilamellar vesicles), the method brings an elegant optimization in one of the polymersome preparation methods however without being a breakthrough in the field.*

Response:

We thank the Reviewer for acknowledging our methodology as an “*elegant optimization*” of nanoprecipitation.

Contrary to the Reviewer’s remark that “*the equilibration loop allows for a good control over the size and shape of polymersomes*”, the equilibration loop plays no actual role in size or shape control. In our system, *size control* was achieved through the use of an annealing loop, which provides heat energy to “nudge” the polymersomes out of their metastable state into lower free energy states. *Shape control*, on the other hand, was enabled by a secondary downstream micromixer—this allowed us to continuously introduce an additive (NaCl) to osmotically deform the polymersomes into their non-spherical, stomatocyte shape. Neither of the two processes have been demonstrated in a continuous downstream fashion as we have reported.

Although we regret that the Reviewer identifies our work as not “*being a breakthrough in the field*”, we will nonetheless attempt to clarify both the novelty and significance of our work through all remaining responses to Reviewer #2 below.

Comment #2: *The Introduction does not contain the real state-of-art in the field regarding the polymersomes production because the well known film rehydration method for polymersome formation and loading with molecules is not presented with its advantages both in terms of polymersomes high yield production and encapsulation efficiency. The Introduction should be improved to present all relevant methods for polymersomes formation and their advantages or still open questions in the field.*

Response:

We believe that we have provided a fairly comprehensive overview of the current state-of-the-art in the Introduction, in particular within the context of nanoprecipitation—the most common bottom-up self-assembly approach in the polymersome field. We acknowledge the existence and importance of the film rehydration method raised by the Reviewer; however, the film rehydration method is a top-down self-assembly approach which bears significant difference to nanoprecipitation (a bottom-up approach) and thus does not fit within the context of our Introduction. The Reviewer also mentions “*encapsulation efficiency*”; however, since we have not performed any encapsulation experiments using our methodology, we do not see any relevance in elaborating on that topic in the Introduction, especially considering its complexity. Below, we outline how our Introduction has been carefully structured to cover a large breadth of information pertaining to polymersome formation via nanoprecipitation, both in batch and in flow, as well as their associated advantages and limitations, and how our methodology opens up the possibility of performing downstream manipulations—processes that would not have been possible with traditional, kinetically trapped polymersomes:

Paragraph 1

- What polymersomes are and how they structurally resemble liposomes
- What physicochemical properties of polymersomes can be modified
- How the above, combined with their ability to load both hydrophilic and hydrophobic materials, has led to widespread applications in drug delivery, synthetic biology and nanoreactor science.

Paragraph 2

- Step-by-step explanation outlining how nanoprecipitation is performed in batch to produce polymersomes
- Explanation of logic behind individual steps in a typical nanoprecipitation process including other important factors to consider

Paragraph 3

- Limitations of batch nanoprecipitation
- Why nanoprecipitation leads to polydisperse polymersomes due to poor mixing efficiency
- Why nanoprecipitation is poorly scalable

Paragraph 4

- How researchers have turned to flow-based systems to negate the effects of mixing in batch

- Examples and advantages/disadvantages of current microfluidic chip-based polymersome formation methods, including:
 - Double emulsion templating using flow-focusing chips
 - Laminar and plugged flow mixing using flow-focusing chips
 - Polymerization-induced self-assembly (PISA) under flow

Paragraph 5

- How miniaturized mixing chambers (micromixers) are superior to flow-focusing chips in both scalability and reproducibility
- How micromixers work by minimizing the timescale of mixing between two incoming streams down to the millisecond regime
- Drawbacks of current polymersome formation methods with micromixers, in particular their ability to only produce kinetically trapped polymersomes
- Explanation as to why downstream processing/manipulation unlocks the full potential of a continuous flow process

Paragraph 6

- Summary, novelty, and rationale behind our work

Comment #3: *There is a confusion the authors include in the Introduction by considering synthetic giant unilamellar vesicles (GUVs) as polymersomes. In the field there is a clear distinction between the vesicles with sizes in the nanometer range (polymersomes) and the vesicles with micrometer sizes (GUVs), similarly to the notions of liposomes and lipid GUVs. This distinction is based both on the methods of production and the properties of the vesicles membrane (stability, curvature, etc). The focus of this manuscript is in preparation of polymersomes. Therefore, the Introduction should be corrected accordingly to avoid misunderstanding and to clearly indicate the relevance of the method the authors propose.*

Response:

The Reviewer's concerns stem from what was written in the Introduction starting Page 2, Line 19. We note that this is the only section in our manuscript where we have discussed micrometre-sized polymersomes that could qualify as so-called *giant unilamellar vesicles (GUVs)*. For clarity, we reproduce the text with the phrase in question underlined below:

"To negate the effects of batch mixing, researchers have turned to flow-based systems such as microfluidics. A reliable microfluidics approach is the double emulsion templating method,²⁸⁻³⁰ which relies on the use of flow-focusing chips to confine and self-assemble block copolymers in the oil phase of water/oil/water (w/o/w) double emulsion droplets. Although the approach generates monodisperse polymersomes with high reproducibility, it is somewhat limited in terms of accessible polymersome size (tens to hundreds of μm), and production scalability because the devices used typically only operate at flow rates of only several $\mu\text{L}/\text{min}$."

We hope it is clear from the above that we have only briefly mentioned micrometre-sized polymersomes from a microfluidic/flow self-assembly context. Given this information, we do not believe it is necessary to explicitly classify or describe polymersomes that are micrometre-sized as *giant unilamellar vesicles (GUVs)*, especially if we also consider the fact that the pioneers of the technique (Prof. David Weitz and his colleagues) themselves referred to such structures simply as "*polymersomes*" in their publications.⁴⁻⁶ Furthermore, from a morphological perspective, there really is no real distinction between sub-micron polymersomes and micrometre-sized polymersomes (e.g., GUVs) other than their sizes, as both "classes" of polymersomes share the same morphology—a bilayer membrane structure and a hollow core. We appreciate that the nomenclature "*giant unilamellar vesicles (GUVs)*" originates from the liposome field; however, we believe the use of this nomenclature should be context dependent. For instance, it may be appropriate when describing a polymersome system that consists of a mixture of giant unilamellar vesicles (GUVs) and multilamellar vesicles (MLVs).

Comment #4: *A serious critical aspect is related to the low mechanic stability of polymersomes obtained by the present method, which downgrade them for any application, as they are less stable even than PEGylated-liposomes. Usually polymersomes stability is of several months, depending on the type of amphiphilic copolymer. Therefore, this method should be significantly improved to allow formation of stable polymersomes for longer periods of time than one week, as this is a real bottle neck for further applications.*

Response:

The Reviewer is certainly correct that our polymersomes are inherently unstable—this is exactly the novelty of our methodology. In the absence of this metastable state, one would not have been able to accomplish downstream manipulation with the level of control that we have demonstrated throughout our manuscript. We would like to further clarify the potential misconception about (i) our polymersomes' metastability in organic solvent/water mixture and (ii) their inherent stability in water below:

Regarding (i) polymersome metastability: Briefly, the term metastability is used to describe a system that exists in an apparent state of equilibrium, when in fact it can transition into a more stable (equilibrium) state if energy is provided to the system. The amount of energy needed for this transition to occur can be quantified as an activation energy barrier (E_A), which in the case of our system, is in the order of $k_B T$ at room temperature (see Figure 2A in main text for proposed free energy diagram). Under ambient conditions, and in organic solvent/water mixture, our polymersomes grow as a result of metastability for ca. 7 days. After this 7-day period, this growth process ceases entirely as the system has transitioned out of its initial metastable state and into an equilibrium state. As we have described in our manuscript, this transition from metastable to equilibrium state is key to the implementation of the downstream annealing setup in Figure 4, which we used to demonstrate size control immediately after self-assembly (and perhaps more importantly, in the very same continuous stream).

Regarding (ii) polymersome stability: As we have discussed on page 9 (line 13 onwards), the metastable state observed in our system can be quenched at any point in time (e.g., during the 7-day growth process or even after the growth has ceased) to trap the system in different kinetically arrested states. This can be done simply by removing the organic solvent from the system (e.g., by extensively dialyzing the polymersomes against water). This is possible owing to high glass transition of polystyrene, PS ($T_{g,PS}$), which is ~ 100 °C. Upon removal of the plasticizing organic solvent, PS chains which constitute the polymersome membrane structure transitions from a dynamic *plasticized* state into a glassy *quenched* state. Once this quenched state has been reached, no further chain rearrangements (and thus no further morphological changes) are possible. In the quenched state, the polymersomes are indefinitely stable unless, of course, some organic solvent is reintroduced into the system to plasticize the PS membrane or if the block copolymer undergoes chemical degradation (which is unlikely in the case of PEO-*b*-PS). Throughout the undertaking of this project, we have observed minimal macroscopic precipitation or sedimentation in all our quenched polymersome samples, some of which have been stored at room temperature for as long as 2 years (although we state here that this stability is not intrinsic to our system and is likely common even for PEO-*b*-PS polymersomes prepared by batch nanoprecipitation). Even if precipitates were present, the samples can simply be filtered through a 0.45 μm polyethersulfone (PES) membrane filter without any deterioration in sample quality as all our polymersome samples are < 0.45 μm in diameter.

With all the above, we dispute the Reviewer's assertion that our polymersomes have "low mechanic(al) stability", "are less stable even than PEGylated-liposomes", and that there "is a real bottle neck for further applications".

Comment #5: *The authors indicate that the method they propose can be used for any type of amphiphilic copolymer. However, they selected as example to test their method for polymersomes formation, PEO-*b*-PS, which has specific properties that cannot be extrapolated for other types of amphiphilic copolymers. In addition, both the chemistry of synthesized block copolymers has been well established and the theoretical background of polymer self-assembly via the solvent switch approach has been previously explored in detail, e.g. by the groups of A. Eisenberg, T.P. Lodge or J.C.M. van Hest. To validate this method for a variety of copolymers, the authors should prove it at least for two different types of copolymers in terms of molecular properties.*

Response:

Comment #6: *The characterization of polymersomes only by DLS (Pag.4 line 25) is not enough to prove the hollow sphere architecture. Static light scattering experiments should be performed and combined to distinguish whether the spherical nanoobjects are polymersomes.*

Response:

For the purpose of this response, we reproduce the statement in question made previously on Page 4, Line 25 underlined and *italicized* below (n.b., this statement is now on Page 4, Line 24 in the revised manuscript), along with the two sentences that preceded it. Also reproduced below is the subsequent paragraph (starting Page 5, Line 3 in the revised manuscript; *italicized*) to help contextualize the statement in question:

"We performed the self-assembly process at 7 different asymmetric flow rates ranging from $Q_{\text{organic}}/Q_{\text{total}} = 0.1-0.7$ (in 0.1 increments). In every case, the product was collected directly into a quartz cuvette and immediately analyzed by dynamic light scattering (DLS). The resulting particle size distributions are shown in Figure 1B. Each sample's intensity-averaged hydrodynamic diameter ($D_{h,\text{intensity}}$) and polydispersity index (PDI) are 25 plotted in Figure 1C. A summary of the DLS data is further provided in Table S1.

All 7 asymmetric flow rates resulted in monomodal particle size distributions with relatively low PDIs of <0.16 (Figure 1B-C and Table S1). At $Q_{\text{organic}}/Q_{\text{total}} \leq 0.2$, minimal changes in particle size were observed. Increments above this value, however, resulted in a linear increase in particle size (see $D_{h,\text{intensity}}$ datapoints for $Q_{\text{organic}}/Q_{\text{total}} = 0.3-0.7$ in Figure 1C). We note here that asymmetric flow rates of $Q_{\text{organic}}/Q_{\text{total}} > 0.7$ were also tested, but these flow conditions did not result in any particle formation because PEO₄₄-b-PS₈₆ remains molecularly dissolved when the organic solvent content exceeds 70 vol%."

As can be seen from the above, we did not make any claims regarding particle morphology using our DLS data. In fact, all our discussions around DLS (data provided in Figure 1B-C and Table S1) was on the effect of asymmetric flow rates ($Q_{\text{organic}}/Q_{\text{total}}$) on particle size ($D_{h,\text{intensity}}$). We specifically noted that (i) particle size does not change when the asymmetric flow rate, $Q_{\text{organic}}/Q_{\text{total}} \leq 0.2$, (ii) particle size increases linearly when the asymmetric flow rate is increased from $Q_{\text{organic}}/Q_{\text{total}} = 0.3-0.7$, and (iii) no particles form beyond $Q_{\text{organic}}/Q_{\text{total}} > 0.7$ because PEO₄₄-b-PS₈₆ is molecularly soluble under those self-assembly conditions. Notice how we were careful in using the term "particle size" as opposed to "micelle size" or "polymersome size" to discuss our DLS data.

We only began making claims on particle morphology starting Page 6, Line 3. For the sake of clarity, we reproduce the entire paragraph below in *italics*, with the claims underlined:

"Next, we used transmission electron microscopy (TEM) to probe particle morphology. Shown in Figure 1D-F are three TEM images of particles produced at $Q_{\text{organic}}/Q_{\text{total}} = 0.2, 0.4$ and 0.6 , respectively. The gradual increase in $Q_{\text{organic}}/Q_{\text{total}}$ generated a morphological transition from micelles (Figure 1D) to a mixed phase of micelles/polymersomes (Figure 1E), and finally to polymersomes (Figure 1F). For clarity, the three accessible morphological phases are highlighted in Figure 1C using different shades of black."

One can see from the above that our claims about particle morphology are solely based on the supporting evidence from our TEM data. We claimed the existence of micelles, a mixed phase of micelles/polymersomes, and polymersomes based on the TEM images provided in Figure 1D, Figure 1E and Figure 1F, respectively.

To summarize, we did not use DLS as a means to prove the hollow structure of our polymersomes. Instead, we relied on TEM (and cryo-TEM throughout many parts of our manuscript) to visualize and confirm our polymersome morphology.

In closing Comment #6, the Reviewer suggested that we perform static light scattering (SLS); however, we do not see any value in doing so since SLS can only (within this context) provide indirect evidence on particle shape based on the shape factor $\rho = R_g/R_h$, and not particle morphology as the Reviewer suggested.⁷ We are confident based on our years of research contributions in the polymersome field (*Chem. Soc. Rev.* **2019**,⁸ *Nat. Commun.* **2017**,⁹ *Chem. Sci.* **2019**,¹⁰ *JACS* **2020**,¹¹ *ACS Nano* **2020**,¹² etc) that TEM and cryo-TEM, both of which we have used to provide direct visual evidence of particle morphology, is a reliable method for confirming polymersome morphology.

Comment #7: Cryo-TEM will be important to give the necessary details of the polymersome membrane that are not clearly visible from the presented TEM micrographs (Fig. 1). Besides, the resolution of all TEM micrographs should be significantly improved. The change in TEM micrographs as stated, “The three accessible morphological phases are highlighted in Figure 1C using different shades of black” is confusing. It is essential to present the raw data, not highlighted images, to avoid biases.

Response:

First of all, we presume that the TEM image in question is Figure 1F specifically (and not Figure 1 as a whole as noted by the Reviewer in Comment #7). We made this presumption because we only provided one TEM image of polymersomes in Figure 1. For the sake of clarity, we reproduce Figure 1F below:

Reproduced Figure 1F. TEM image of polymersomes obtained at $Q_{\text{organic}}/Q_{\text{total}} = 0.6$. This sample was analyzed in their respective organic solvent/water mixtures. Shown on the right (and highlighted red) is a magnified region of the same TEM image where the morphology of individual polymersomes can clearly be seen.

We struggle to understand why the Reviewer isn't convinced that the particles shown above in Reproduced Figure 1F are polymersomes considering how well-resolved the membrane structure of individual particles are in the image. In their comment, the Reviewer further suggested that “cryo-TEM will be important to give the necessary details of the polymersome membrane”.

In response, we point out that there was in fact a reason why we could not measure cryo-TEM for the sample in Figure 1F. To explain this, we refer to Page 6, Lines 1-2, where we stated that the sample in Figure 1F was “analyzed in their respective organic solvent/water mixtures”. For clarify, the “organic solvent/water mixture” in this sample consists of 60 vol% of organic solvent (20% THF/dioxane) and 40 vol% of water. Due to the large amount of organic solvent present in this sample (and its miscibility with liquid ethane, which we use for sample vitrification), we are unable to obtain a vitrified sample that was good enough for cryo-TEM imaging. That said, it is well established in the literature that PEO-*b*-PS can form glassy polymersomes that retain their structure in the dry state—well enough to be imaged by dry state TEM.^{2,13,14} TEM imaging of PEO-*b*-PS polymersomes generally becomes challenging (i) if the polymersomes studied are e.g., ≥ 500 nm in diameter, because at such sizes, they tend to buckle or collapse when dried because their (~20 nm-thin) membrane structure can no longer support the overall diameter of the structure, or (ii) if the polymersomes have non-spherical shapes that are difficult to properly characterize in the dry-state. In such cases, cryo-TEM becomes a necessary tool to confirm polymersome morphology in a pristine, frozen-hydrated state.¹⁵

We further point out that we have in fact provided a fair amount of cryo-TEM data throughout the manuscript (wherever feasible and necessary) to confirm our polymersomes' morphology and/or non-spherical shapes. These cryo-TEM images can be found in Figure 2D, Figure 2E, Figure 2H(ii), Figure 2H(iii), and Figure 5C.

In closing Comment #7, the Reviewer suggests that (i) the “resolution of all TEM micrographs should be significantly improved”, (ii) our statement that “The three accessible morphological phases are highlighted in Figure 1C using different shades of black” is confusing, and that “it is essential to present the raw data, not highlighted images, to avoid biases.

In response to (i) issue with TEM image resolution: All TEM images presented throughout our manuscript were acquired at the highest possible resolution ($\sim 4112 \times 3008$ pixels) and saved at a file size between 35-40 megabytes (MB). Although we disagree that the resolution of our images needs to be improved, we point out that further improvements to image resolution are not possible due to the limitations of the camera (EMESIS Phurona) on our microscope.

In response to (ii) issue with our statement: We begin by clarifying that the statement in question was made on Page 5, Line 13. It was provided to supplement the figure caption of Figure 1C, which we reproduce as follows:

“(C) Intensity-averaged hydrodynamic diameters ($D_{h, intensity}$) and polydispersity indices (PDI) derived from the data shown in B. The different shades of black in C depict a pseudo-phase diagram.”

As can be seen from the above, the figure caption advises readers that the DLS data in Figure 1C has different shades of black to depict a pseudo-phase diagram and to help readers visually identify the different morphologies that can be accessed. The Reviewer appears to have misunderstood our statement since they somehow suggested that *“it is essential to present the raw data, not highlighted images, to avoid biases”*. We clarify that none of our TEM images have been “highlighted” to potentially generate bias—the only thing that has been highlighted was the DLS data in Figure 1C to depict a pseudo-phase diagram.

We have nevertheless amended the phrases *“different shades of black”* with *“different shades of gray”* in our manuscript to hopefully avoid future confusion.

Comment #8: *Figures 2 D and E regarding TEM micrographs of polymersomes are different from what we expect to have when hollow sphere architecture is present; the spherical nanoobjects have a darker core that is not specific for polymersomes. Therefore it is not clear that the nanoobjects in these figures are polymersomes. Cryo-TEM will elucidate this issue.*

Response:

We begin our response by reproducing Figures 2D and 2E below:

Reproduced Figure 2. TEM images of (D) pristine polymersomes quenched on day 0 (immediately after continuous flow self-assembly) and (E) aged polymersomes quenched after 14 days of aging. Corresponding cryo-TEM images are shown inset in D and E.

We presume that the Reviewer is referring to the inset cryo-TEM images in Figure 2D and 2E (and not Figure 2D and 2E as a whole) as the inset images are the only images where our polymersomes “*have a darker core*” that is, according to the Reviewer, “*not specific for polymersomes*”.

We argue that this “dark core” that the Reviewer raises as an issue is in fact a cryo-TEM imaging artifact that has long been known to exist in the vesicle literature. This imaging artifact has been reported as early as in 2000 by Almgren *et al.*¹⁶ For clarity, we reproduce a cryo-TEM image of liposomes with “dark cores” reported in the cited work, along with the figure caption that was published alongside the cryo-TEM image.

Fig. 6. Large liposomes protruding out of the vitrified film gives an image that is darkest in the central, thickest part. Compare the drawing in Fig. 2.

Figure R2. A cryo-TEM image of liposomes reported by Almgren *et al.*¹⁶ with “dark cores” similar to what we have presented in Figures 2D and 2E of our manuscript. Note that the original figure caption has been reproduced below the cryo-TEM image for clarity.

As pointed out by Almgren *et al.*¹⁶ in their figure caption, “*the large liposomes protruding out of the vitrified film gives an image that is darkest in the central, thickest part*”. To help the Reviewer understand this statement, we provide a schematic below in Figure R3 to illustrate the phenomenon:

Figure R3. A simplified schematic illustrating why polymersomes (or vesicles in general) sometimes have “dark cores” when visualized under cryo-TEM.

We further provide four literature examples^{15,17–19} below in Figure R4 where polymersomes have been reported with “dark cores” under cryo-TEM.

Figure 2a-f from Rikken *et al. Nat. Commun.* **2016**, 7, 12606

Figure 2A-C from Ridolfo *et al. Polym. Chem.* **2020**, 11, 2775-2780

Figure 2B-C from Rijpkema *et al. Biomacromolecules* **2020**, 21, 1853-1864

Figure 2A from Abdelmohsen *et al. JACS* **2016**, 138, 9353-9356

Figure R4. Cryo-TEM images of polymersomes from 4 different references. In each example provided, every polymersome can be seen to exhibit the same “dark core”, which the Reviewer has claimed to be non-specific to polymersomes.

Finally, in closing Comment #8, the Reviewer suggested that the use of cryo-TEM could potentially “elucidate this issue” (n.b., “this issue” implies the observation of the dark cores). We clarify here that the images shown inset in Figures 2E and 2D, which the Reviewer raised issues with, are in fact cryo-TEM images.

Comment #9: Figures 4 D-F indicate aggregation of the spherical nanoobjects, which is a severe limitation for further applications. How the aggregation can be avoided?

Response:

We understand that the density of particles in these TEM images gives the impression that our polymersomes are aggregated. However, our DLS data presented in Figures 4B-C very clearly indicates the absence of any aggregation phenomena. Every polymersome sample that was annealed between 20-70 °C (including those

whose TEM images were provided in Figure 4D-F) gave monomodal size distributions on DLS with PDIs ≤ 0.10 , indicating sample uniformity without the presence of any aggregates. See Table S7 for exact $D_{h,intensity}$ and PDI values for each sample. We prefer to retain the same TEM images in Figure 4D-F as they provide an overview of >30 polymersomes per image as opposed to just a select few. Furthermore, if we were to dilute these samples and replace the TEM images to show only a few scattered polymersomes per image, the size differences may not as immediately clear to readers.

Reviewer #3 (Remarks to the Author):

In this manuscript, the authors reported a continuous flow methodology capable of producing near-monodisperse polymersomes at scale (≥ 3 g/h). They also demonstrated downstream processes (thermal annealing and/or secondary micro-mixing) to manipulate polymersome size (with sub-40 nm precision) and/or polymersome shape. This work is meaningful for the production of near-monodisperse polymersomes at scale, as well as the control of polymersomes properties under flow conditions. I would recommend its publication on NC after major revision.

Comment #1: *The authors claimed that they specifically adjusted the salinity of the polymersome solution to 50 mM NaCl, followed by dialysis to remove the organic solvents. Why choose 50 mM NaCl?*

Response:

We apologize for the lack of clarity.

We specifically chose 50 mM NaCl as the osmotic additive because this particular salt and concentration is commonly used to induce osmotic pressures that are sufficiently strong to cause polymersomes to deform into non-spherical shapes. Previous systematic studies^{20–22} have demonstrated that NaCl concentrations below 50 mM generally result in only partial deformation, while NaCl concentrations exceeding 50 mM do not produce any noticeable effects beyond complete shape transformation.

We have amended the text on Page 11, Line 6 and added 3 references to clarify this:

“We specifically adjusted the salinity of the polymersome solution to 50 mM NaCl (a concentration regularly used to deform polymersomes by osmotic pressure),^{47–49} followed by dialysis to remove the organic solvents.”

Comment #2: *At the highest total flow rate ($Q_{total} = 8$ mL/min), a low PDI of 0.045 ± 0.015 was obtained, indicating near-monodisperse polymersomes. The monodispersity is very interesting, and the authors should investigate the formation mechanism of the monodispersity.*

Response:

We begin our response by clarifying that the Reviewer is referring to our data in Figure 3B, which demonstrates that an increase in total flow rate (Q_{total}) leads to a significant reduction in polydispersity (PDI). At the highest total flow rate that was tested ($Q_{total} = 8$ mL/min), we observed a particularly low PDI of 0.045 ± 0.015 , indicating the presence of near-monodisperse polymersomes.

One plausible explanation for the observed decrease in PDI with increasing total flow rate (Q_{total}) is the transition from transient to turbulent flow regime. In the lower range of Q_{total} (0.5–4 mL/min), we observed noticeable improvements in PDI as Q_{total} increased. Given that the extent of improvement in PDI was very pronounced in this range, these improvements are likely attributed to micromixing occurring in the transient flow regime.²³

At the highest Q_{total} tested (8 mL/min), the remarkably low PDI observed (0.045 ± 0.015) suggests that the micromixing process is most efficient in this flow regime, thereby enhancing the uniformity of the self-assembly process. Considering how there were minimal improvements in PDI beyond this, we postulate that a complete transition from transient to turbulent flow regime occurs at $Q_{total} = 8$ mL/min.

While these observations offer valuable insights, it is important to acknowledge that our understanding of the exact flow regimes involved is currently limited, and a comprehensive understanding would require extensive computational studies. This is particularly the case because the micromixer employed in our study is a commercially available product, and thus we lack detailed knowledge of its internal geometry and specifications to properly calculate Reynolds number (Re), a dimensionless parameter that is used to characterize a fluid flow profile. We would have to engineer a custom micromixer in order to conduct such computational investigations. While we acknowledge the importance of this, we want to clarify that such endeavors are beyond the scope of our current work, which revolves around polymersome metastability and its implications in continuous flow manufacturing. We will consider this in our future research directions, but at this stage, we wish to avoid speculating on flow regimes/profiles in the main text, and prefer to retain our original general explanation attributing the improvements in PDI to an increase in flow turbulence in the micromixer (which we believe is reasonable since total flow rate (Q_{total}) is the only parameter changed in the experiment):

“We attribute the decreasing size and PDI trends at higher Q_{total} to an increase in flow turbulence during micromixing (Figure S5). The effect of Q_{total} , however, diminishes beyond $Q_{total} \geq 8$ mL/min as flow turbulence can no longer be improved beyond the limitations imposed by the geometry of the micromixer.”

Comment #3: A production rate of ≥ 3 g of polymersomes/hour was achieved. Further increasing $C_{polymer}$ and Q_{total} , the production rate can be improved. Why the authors did not pursue this?

Response:

The Reviewer poses an excellent question here.

Indeed, the production can be improved by simultaneously increasing polymer concentration ($C_{polymer}$) and the total flow rate (Q_{total}). By augmenting both of these variables in parallel, one can effectively enhance the production rate beyond 3 g/h. However, we refrained from exploring conditions surpassing $C_{polymer} = 9$ mg/mL and $Q_{total} = 8$ mL/min (data shown in Figure 3D) due to the need of more than 72 mg of PEO₄₄-*b*-PS₈₆ per minute to conduct such experiments.

To put things into a perspective, let's consider an experimental scenario employing the following self-assembly conditions: $C_{polymer} = 20$ mg/mL, $Q_{total} = 20$ mL/min (and $Q_{organic}/Q_{total} = 0.7$ to target polymersomes). Under these conditions, the production rate could reach as high as 16.8 g/h. However, one needs to appreciate that a mere 5-minute experiment under such conditions would necessitate 1.4 g of PEO₄₄-*b*-PS₈₆, which accounts for approximately three-fourths of the total amount of PEO₄₄-*b*-PS₈₆ we have synthesized for the entire project (see experimental section in SI).

We clarify here that a discussion on this had already been provided in the main text starting Page 13, Line 9:

*“What is important to recognize here is that these flow conditions equate to a production rate of ≥ 3 g of polymersomes/hour, far exceeding the capabilities of typical batch self-assembly processes. The production rate demonstrated herein can undoubtedly be improved by further increasing $C_{polymer}$ and Q_{total} , but we did not pursue this as such experiments would require >72 mg of PEO₄₄-*b*-PS₈₆/minute to conduct.”*

Comment #4: The authors also manipulated polymersome shape by expanding the flow setup to include a cooling loop and a secondary micromixer connected to another syringe pump. It was crucial to introduce only a small amount of concentrated NaCl solution, as this ensures minimal deviations in solvent quality after micromixing, thus preventing any morphological deviations. How to precisely control a small amount of concentrated NaCl solution? What is the exact value and exact quantity? Any preliminary data/experiment?

Response:

The Reviewer accurately notes that the flow setup had to be expanded to include a cooling loop and a secondary micromixer (connected to a third syringe pump) to manipulate polymersome shape.

To answer the Reviewer's query, the small amount of concentrated NaCl solution ($C_{NaCl} = 5.05$ M) was introduced into the system through the inlet of the secondary micromixer. A photograph of the entire setup

can be found in Figure S9 in the SI. The amount of 5.0 M NaCl solution added was precisely controlled by the syringe pump connected to the secondary micromixer.

We clarify that the information mentioned above has been provided in the main text starting Page 17, Line 11:

“The secondary micromixer, which is placed downstream of the cooling loop (Figure 5A), serves as a junction for the introduction of an additive (NaCl solution) needed to osmotically deform the annealed/grown polymersomes. In a typical experiment, we would generate a salinity change of 50 mM NaCl by micromixing the annealed/grown polymersome solution with a concentrated NaCl solution (5.05 M) at a flow rate of 4 mL/min and 0.04 mL/min, respectively. We found it crucial to introduce only a small amount of concentrated NaCl solution (as opposed to larger volumes of diluted NaCl solution) as this ensures minimal deviations in solvent quality after micromixing, thus preventing any morphological deviations beyond the intended shape transformation process.”

As shown above, we specified that the 5.05 M NaCl solution was introduced into the system at a flow rate of 0.04 mL/min (n.b., these experimental conditions were also provided in the caption of Figure 5C and in the Experimental Section on Page 19 of the SI). In other words, our statement implies that, in every 1 minute, the syringe pump dispenses a total of 40 μ L of 5.05 M NaCl solution into the incoming stream of annealed/grown polymersomes, which is conversely flowed at 4 mL/min. Accounting mutual dilution when these two solutions are mixed, the final salinity is thus $c_{\text{NaCl,final}} = [(C_{\text{NaCl}} \times V_{\text{NaCl}}) / (V_{\text{NaCl}} + V_{\text{polymersome}})] = [(5.05 \text{ M} \times 0.040 \text{ mL}) / (0.040 \text{ mL} + 4 \text{ mL})] = 50 \text{ mM}$. Based on the same information, we can also calculate the extent of dilution of the incoming stream using the following formula: $\% \text{dilution} = [1 - (Q_{\text{total,original}} / Q_{\text{total,original+NaCl}}) \times 100\%] = [1 - (4 / 4.04) \text{ mL/min} \times 100\%] = 1\%$, and hence our claims of “minimal deviations in solvent quality after micromixing”.

Rebuttal References

1. Markwalter, C. E. *et al.* Polymeric Nanocarrier Formulations of Biologics Using Inverse Flash NanoPrecipitation. *AAPS J.* **22**, 1–16 (2020).
2. Kim, K. T. *et al.* Polymersome stomatocytes: controlled shape transformation in polymer vesicles. *J. Am. Chem. Soc.* **132**, 12522–12524 (2010).
3. Matoori, S. *et al.* An Investigation of PS-b-PEO Polymersomes for the Oral Treatment and Diagnosis of Hyperammonemia. *Small* **15**, 1–13 (2019).
4. Ho, C. S., Kim, J. W. & Weitz, D. A. Microfluidic fabrication of monodisperse biocompatible and biodegradable polymersomes with controlled permeability. *J. Am. Chem. Soc.* **130**, 9543–9549 (2008).
5. Shum, H. C., Zhao, Y. J., Kim, S. H. & Weitz, D. A. Multicompartment polymersomes from double emulsions. *Angew. Chem. Int. Ed.* **50**, 1648–1651 (2011).
6. Amstad, E., Kim, S. H. & Weitz, D. A. Photo- and thermoresponsive polymersomes for triggered release. *Angew. Chem. Int. Ed.* **51**, 12499–12503 (2012).
7. Abdelmohsen, L. K. E. A., Rikken, R. S. M., Christianen, P. C. M., van Hest, J. C. M. & Wilson, D. A. Shape characterization of polymersome morphologies via light scattering techniques. *Polymer* **107**, 445–449 (2016).
8. Wong, C. K., Stenzel, M. H. & Thordarson, P. Non-spherical polymersomes: Formation and characterization. *Chem. Soc. Rev.* **48**, 4019–4035 (2019).
9. Wong, C. K., Mason, A. F., Stenzel, M. H. & Thordarson, P. Formation of non-spherical polymersomes driven by hydrophobic directional aromatic perylene interactions. *Nat. Commun.* **8**, 1240 (2017).
10. Wong, C. K. *et al.* Faceted polymersomes: a sphere-to-polyhedron shape transformation. *Chem. Sci.* **10**, 2725–2731 (2019).
11. Wong, C. K. *et al.* Vesicular Polymer Hexosomes Exhibit Topological Defects. *J. Am. Chem. Soc.* **142**, 10989–10995 (2020).
12. Gröschel, T. I., Wong, C. K., Haataja, J. S., Dias, M. A. & Gröschel, A. H. Direct observation of

- topological defects in striped block copolymer discs and polymersomes. *ACS Nano* **14**, 4829–4838 (2020).
13. Wilson, D. A., Nolte, R. J. M. & van Hest, J. C. M. Autonomous movement of platinum-loaded stomatocytes. *Nat. Chem.* **4**, 268–74 (2012).
 14. Van Rhee, P. G. *et al.* Polymersome magneto-valves for reversible capture and release of nanoparticles. *Nat. Commun.* **5**, 1–8 (2014).
 15. Rikken, R. S. M. *et al.* Shaping polymersomes into predictable morphologies via out-of-equilibrium self-assembly. *Nat. Commun.* **7**, 12606 (2016).
 16. Almgren, M., Edwards, K. & Karlsson, G. Cryo transmission electron microscopy of liposomes and related structures. *Colloids Surfaces A Physicochem. Eng. Asp.* **174**, 3–21 (2000).
 17. Abdelmohsen, L. K. E. A. *et al.* Formation of well-defined, functional nanotubes via osmotically induced shape transformation of biodegradable polymersomes. *J. Am. Chem. Soc.* **138**, 9353–9356 (2016).
 18. Ridolfo, R., Williams, D. S. & Van Hest, J. C. M. Influence of surface charge on the formulation of elongated PEG-*B*-PDLLA nanoparticles. *Polym. Chem.* **11**, 2775–2780 (2020).
 19. Rijpkema, S. J. *et al.* Modular Approach to the Functionalization of Polymersomes. *Biomacromolecules* **21**, 1853–1864 (2020).
 20. Pijpers, I. A. B., Abdelmohsen, L. K. E. A., Williams, D. S. & Van Hest, J. C. M. Morphology under Control: Engineering Biodegradable Stomatocytes. *ACS Macro Lett.* **6**, 1217–1222 (2017).
 21. Wauters, A. C. *et al.* Development of Morphologically Discrete PEG–PDLLA Nanotubes for Precision Nanomedicine. *Biomacromolecules* **20**, 177–183 (2019).
 22. Men, Y., Li, W., Lebleu, C., Sun, J. & Wilson, D. A. Tailoring Polymersome Shape Using the Hofmeister Effect. *Biomacromolecules* **21**, 89–94 (2020).
 23. Plutschack, M. B., Pieber, B., Gilmore, K. & Seeberger, P. H. The Hitchhiker’s Guide to Flow Chemistry. *Chem. Rev.* **117**, 11796–11893 (2017).

Editorial note: Reviewer 2 was unable to look over the responses to the comments, and therefore Reviewer 1 assessed the responses to these comments.

REVIEWERS' COMMENTS

Reviewer #1 (Remarks to the Author):

My comments have been thoroughly addressed. I support acceptance of this publication and applaud the authors on their hard work.

I also believe that the comments of Reviewer 2 were thoroughly addressed.

Reviewer #3 (Remarks to the Author):

After revision, the authors did quite a few works to improve the quality of the paper. I would recommend this manuscript for this format to be accepted and published.

REVIEWER COMMENTS

Reviewer #1 (Remarks to the Author):

My comments have been thoroughly addressed. I support acceptance of this publication and applaud the authors on their hard work. I also believe that the comments of Reviewer 2 were thoroughly addressed.

Response:

We appreciate the time and support that the Reviewer has dedicated to the reviewing process.

Reviewer #3 (Remarks to the Author):

After revision, the authors did quite a few works to improve the quality of the paper. I would recommend this manuscript for this format to be accepted and published.

Response:

We thank the Reviewer for their time and for endorsing publication of our manuscript.